# FANS: A Flatness-Aware Network Structure for Generalization in Offline Reinforcement Learning

**Da Wang**[1], **Yi Ma**[1]\*, **Ting Guo**[2], **Hongyao Tang**[3], **Wei Wei**[1], **Jiye Liang**[1]

[1]School of Computer and Information Technology, Shanxi University,
{wanda, mayi, weiwei, ljy}@sxu.edu.cn
[2]Data Science and Technology, North University of China, 20240017@nuc.edu.cn
[3]College of Intelligence and Computing, Tianjin University, bluecontra@tju.edu.cn

## Abstract

Offline reinforcement learning (RL) aims to learn optimal policies from static datasets while enhancing generalization to out-of-distribution (OOD) data. To mitigate overfitting to suboptimal behaviors in offline datasets, existing methods often relax constraints on policy and data or extract informative patterns through data-driven techniques. However, there has been limited exploration into structurally guiding the optimization process toward flatter regions of the solution space that offer better generalization. Motivated by this observation, we present *FANS*, a generalization-oriented structured network framework that promotes flatter and robust policy learning by guiding the optimization trajectory through modular architectural design. FANS comprises four key components: (1) Residual Blocks, which facilitate compact and expressive representations; (2) Gaussian Activation, which promotes smoother gradients; (3) Layer Normalization, which mitigates overfitting; and (4) Ensemble Modeling, which reduces estimation variance. By integrating FANS into a standard actor-critic framework, we highlight that this remarkably simple architecture achieves superior performance across various tasks compared to many existing advanced methods. Moreover, we validate the effectiveness of FANS in mitigating overestimation and promoting generalization, demonstrating the promising potential of architectural design in advancing offline RL.

## 1 Introduction

Offline reinforcement learning (RL) [1] focuses on learning policies from fixed, pre-collected datasets without access to online interactions with the environment. A fundamental challenge in this paradigm is the distributional shift between the offline dataset and the true environment dynamics encountered during deployment. This discrepancy often leads to unreliable generalization, particularly in out-of-distribution (OOD) regions, where value functions tend to exhibit overestimation [2]. To mitigate this, prior work has introduced approaches such as policy constraints [3, 4, 5, 6, 7], value function regularization [8, 9, 10, 11], and uncertainty estimation [12, 13, 14]. These methods aim to constrain the learning process, ensuring that the learned policy avoids making unreliable decisions in regions of state space with overestimated values, thereby enabling the model to derive more robust and effective policies from offline datasets.

In recent years, generalization has emerged as a central focus in offline RL, driven by the need to ensure reliable policy performance beyond the narrow support of the training data. To enhance generalization, existing methods typically relax constraints on the dataset [15], behavior policy [7], or support [5, 16] during the learning process. In addition, some data-driven approaches [17, 18] formulate generalization as a domain adaptation problem, treating the discrepancy between

---

\*Corresponding author.

training and deployment distributions as a shift between source and target domains. Despite these advances, relatively little effort has been devoted to exploring how neural network architectures might influence generalization. While most efforts in offline RL have centered around data and optimization constraints, the role of model architecture in promoting generalization remains relatively underexplored. Yet, architectural design can significantly influence the learning dynamics and the generalization behavior of the resulting policy.

Recent studies have demonstrated that specific architectural choices − such as residual connections [19], layer normalization [20, 21], and smooth activation functions − can implicitly bias optimization toward flatter regions of the loss landscape, which are often associated with improved generalization. While state-of-the-art architectures like SimBa [22] have achieved remarkable scalability in deep RL, the most suitable architectural design for offline RL remains an open question. Given the unique challenges of offline RL − particularly distributional shift and limited data coverage, which often lead to sharp, overfitted solutions − incorporating architectural inductive biases that promote flatter minima represents a promising and complementary direction for improving generalization. Motivated by these insights, we propose **FANS** (Flatness-Aware Network Structure), a structural framework specifically designed to enhance generalization in offline RL. FANS comprises four key modules, each tailored to encourage flatter solutions and improve stability:

1. **Residual Blocks**: Facilitate learning simple, clean mappings, enabling smoother gradient flow and mitigating the risk of overfitting to noisy or sparse data points.

2. **Gaussian Activation Function**: Replace traditional piecewise-linear activation (e.g., ReLU) with smoother functions, promoting continuous gradients and flatter local loss landscapes.

3. **Layer Normalization**: Regularizes feature distributions across layers, helping to stabilize optimization dynamics and preventing sharp activations that could lead to overfitting.

4. **Model Ensemble**: Aggregates multiple models to reduce variance and bias, ensuring the learned policy is not overly sensitive to specific trajectories or regions in the training data.

Together, these components systematically bias the optimization process toward solutions located in flatter regions of the loss landscape, thereby enhancing the model's ability to generalize to unseen OOD data while maintaining strong performance in well-covered regions. Importantly, by introducing the FANS framework into a standard Actor-Critic (AC) architecture without modifying the objective function, we observe substantial performance improvements. This underscores the effectiveness of architectural design in addressing the unique generalization challenges inherent to offline RL.

**In summary, our contribution is three-fold:**

1. We propose a structured network design framework for offline RL that integrates residual blocks, Gaussian activation function, layer normalization, and ensemble techniques to enhance generalization.

2. We validate the effectiveness of the proposed framework across multiple offline RL tasks, highlighting that our remarkably simple architecture leads to substantial performance gains.

3. We conduct a detailed analysis to elucidate how the FANS framework facilitates smoother optimization, reduces variance, and mitigates overfitting, thereby achieving significant improvements in OOD generalization performance.

## 2 Preliminary and Related Work

### 2.1 Offline RL

Offline RL, also referred to as batch RL, aims to learn an optimal policy solely from a fixed dataset $\mathcal{D} = \{(s, a, r, s')\}$, collected by one or more behavior policies, without further interaction with the environment. The learning problem is typically formulated within the Markov Decision Process framework, defined by a tuple $(\mathcal{S}, \mathcal{A}, \mathcal{P}, r, \gamma)$, where $\mathcal{S}$ is the state space, $\mathcal{A}$ is the action space, $\mathcal{P}(s'|s, a)$ is the transition probability, $r(s, a)$ is the reward function, and $\gamma \in (0, 1)$ is the discount

factor. The objective of offline RL is to learn a policy $\pi(a|s)$ that maximizes the expected cumulative reward over the data distribution induced by the behavior policy, formulated as

$$J(\pi) := \mathbb{E}_{(s,a)\sim\mathcal{D}}\left[Q^\pi(s,a)\right] = \mathbb{E}_{(s,a)\sim\mathcal{D}}\left[\mathbb{E}_\pi\left[\sum_{t=0}^\infty \gamma^t r(s_t,a_t) \mid s_0=s, a_0=a\right]\right], \quad (1)$$

where $Q^\pi(s,a)$ denotes the expected discounted return starting from state $s$, taking action $a$, and following policy $\pi$ thereafter. Unlike online RL [23], where the agent iteratively interacts with the environment to refine its policy, offline RL faces the fundamental challenge of distributional shift: the learned policy $\pi$ may generate actions and state-action pairs that are OOD concerning the dataset $\mathcal{D}$. This can lead to overly optimistic value function estimates, hindering policy performance.

**Generalization in offline RL.** As attention to distribution shift increases in offline RL, various methods have been proposed to address the challenge. Among them, traditional conservative approaches typically constrain learned policies to remain within the dataset's support, aiming to suppress the generation of OOD actions. Representative approaches include explicit policy constraints [4, 3, 5, 6, 24], value function penalization [8, 10, 25, 26], uncertainty quantification techniques [12, 13, 14], and the integration of imitation learning [27, 9]. In most cases, these approaches favor conservative strategies to ensure the reliability and safety of the policies learned from limited offline data.

Building upon foundations laid by prior research, subsequent works have concentrated on refining methods to alleviate excessive conservatism, thereby enhancing generalization capabilities. For instance, MCQ [15] actively trains on OOD actions by constructing pseudo target values. SPOT [5] explicitly models the behavior policy's support using a VAE-based density estimator and introduces a simple, pluggable density-based regularization to effectively constrain offline RL policies. DOGE [28] leverages a learned distance function to guide policy learning beyond the data distribution. TSRL [29] exploits time-reversal symmetry in dynamics to improve representation learning and reliability estimation, enabling data-efficient and generalizable offline RL from small datasets. POR [30] inherits the training stability of imitation-style methods while still allowing logical OOD generalization. STR [16] performs trust region policy optimization within the support of the behavior policy. Additionally, some data-driven approaches like PRDC [7] have found that regularizing policies toward the nearest state-action pairs is more effective, enabling the learned policy to select actions outside the dataset for a given state. Other studies [17, 18] innovatively model the OOD generalization challenge in offline RL from a distribution adaptation perspective. Orthogonal to existing methods, our approach leverages minimal architectural modifications to achieve impressive performance, offering a structural perspective largely overlooked in prior work.

## 2.2 Network Architecture Design and the Flatness of Optimization Landscapes

Early deep RL largely overlooked network architecture design, often relying on simple MLPs [31], which, under RL's non-stationarity and trial-and-error learning, exhibited optimization pathologies such as capacity loss [32], primacy bias [33], and plasticity loss [34]. These issues worsen as model scale increases [35], highlighting the urgent need for architectural innovations to alleviate training pathologies and enhance generalization.

Recent research has begun to address these challenges by leveraging network architecture design to steer optimization toward flatter minima in the loss landscape. At the macro level, techniques such as **normalization** and **residual connections** have demonstrated significant benefits. Methods like spectral normalization [36], batch normalization [37], and the widely used layer normalization [20, 38] effectively control gradient magnitudes and stabilize parameter updates. This regulation helps smooth the loss landscape by preventing excessively sharp or irregular surfaces, thereby facilitating more stable training dynamics and faster convergence. SEEM [39] identifies a self-excitation mechanism causing Q-value divergence in offline RL and shows that it can be effectively suppressed by LayerNorm. Additionally, BRO [40] and SimBa [22] demonstrate that incorporating residual blocks significantly improves training robustness and performance, with SimBa further enhancing stability by adding observation normalization layers, establishing it as a widely adopted state-of-the-art architecture in deep RL.

At the micro level, the choice of **activation function** also critically influences the geometry of the loss landscape. In particular, Gaussian-based activations such as GELU [41], with their smooth curvature and continuous higher-order derivatives, have been theoretically and empirically shown to encourage

optimization trajectories toward flatter regions, thereby enhancing generalization performance [42]. Compared to traditional piecewise linear functions like ReLU, these activations maintain nonlinear expressivity while mitigating gradient discontinuities that can destabilize training. Together, these micro-level designs complement macro-level architectural choices, jointly shaping more favorable optimization paths and improving model generalization.

## 3 FANS framework: Flatness-Aware Network Structure

To address the distributional shift challenge in offline RL −particularly the lack of generalization when encountering OOD states − we propose a structure-enhanced network framework, FANS. Our framework leverages a structurally guided design that incorporates several optimization-aware architectural components to steer the training dynamics toward flatter regions of the loss landscape, thereby enhancing generalization.

Specifically, we employ residual connections, layer normalization, a smooth Gaussian activation function, and model ensembling. An overview of the FANS architecture is illustrated in Figure 1.

Given an input vector $\mathbf{x} \in \mathbb{R}^d$, the standardization process first computes the mean $\boldsymbol{\mu} \in \mathbb{R}^d$ and standard deviation $\boldsymbol{\sigma} \in \mathbb{R}^d$ across the dataset as

$$\boldsymbol{\mu} = \frac{1}{N} \sum_{i=1}^{N} \mathbf{x}_i, \quad \boldsymbol{\sigma} = \sqrt{\frac{1}{N} \sum_{i=1}^{N} (\mathbf{x}_i - \boldsymbol{\mu})^2 + \varepsilon},$$

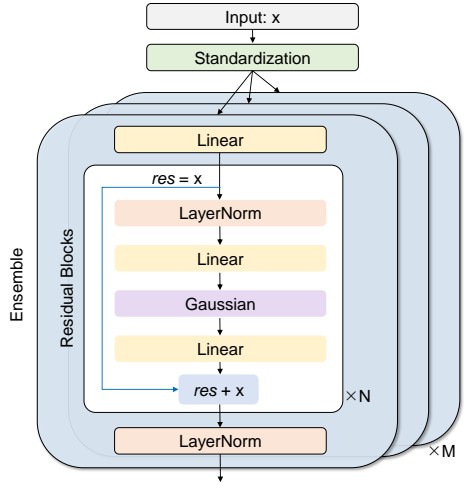

Figure 1: **FANS architecture.**

where $\{\mathbf{x}_i\}_{i=1}^{N}$ denotes the set of input vectors and $\varepsilon$ is a small constant for numerical stability. The standardized output is then obtained by $\mathbf{x} = \frac{\mathbf{x} - \boldsymbol{\mu}}{\boldsymbol{\sigma}}$. This transformation ensures that each input dimension has zero mean and unit variance, which improves optimization stability and prevents scale-sensitive biases in the learning process.

**Residual Block.** Each residual block in FANS is designed to encourage smoother optimization and more stable representation learning, addressing the sharp loss landscapes and overfitting risks inherent in offline RL. The architecture follows a pre-activation residual design, incorporating normalization, Gaussian nonlinearity, and linear projection within the skip-connected block. The residual path is shown in Table 1, where $\mathbf{W}_1, \mathbf{W}_2 \in \mathbb{R}^{d \times d}$ and $\mathbf{b}_1, \mathbf{b}_2 \in \mathbb{R}^d$ are learnable parameters.

Table 1: Residual Block Architecture in FANS. Each step operates on a hidden vector of dimension $d$.

| Step | Operation | Equation | Description |
|------|-----------|----------|-------------|
| (1) | Residual Save | $\mathbf{res} = \mathbf{x}$ | Store input for residual connection |
| (2) | LayerNorm | $\mathbf{h}_1 = \text{LayerNorm}(\mathbf{x})$ | Normalize input across features |
| (3) | Linear Layer 1 | $\mathbf{h}_2 = \mathbf{W}_1 \mathbf{h}_1 + \mathbf{b}_1$ | First linear transformation |
| (4) | Gaussian Activation | $\mathbf{h}_3 = \exp(-\mathbf{h}_2^2)$ | Smooth, non-monotonic nonlinearity |
| (5) | Linear Layer 2 | $\mathbf{h}_4 = \mathbf{W}_2 \mathbf{h}_3 + \mathbf{b}_2$ | Second linear transformation |
| (6) | Residual Add | $\mathbf{y} = \mathbf{res} + \mathbf{h}_4$ | Residual connection output |

This structure preserves the identity mapping through residual addition, stabilizing gradient propagation and promoting generalization through implicit regularization.

**Gaussian Activation Function.** The Gaussian activation function used in each residual block is:

$$\phi(u) = \exp(-u^2), \tag{2}$$

where $u$ is the output of a linear transformation. Unlike ReLU or other monotonic activations, the Gaussian function is smooth and bell-shaped, with bounded output in $(0, 1]$. Its non-monotonicity enables richer functional representations and encourages localized sensitivity.

**Layer Normalization.** To stabilize training and enhance generalization, we adopt *post-layer normalization* after the final residual block. Formally, for an intermediate output $\mathbf{y}$, we compute:

$$\mathbf{z} = \text{LayerNorm}(\mathbf{y}). \tag{3}$$

Additionally, this operation reduces sensitivity to feature scaling and enhances the smoothness of the optimization landscape, which is particularly beneficial when combined with activation functions such as the Gaussian function. Finally, it ensures consistent feature magnitudes across modules, which is important when the output is passed into a common prediction layer. Following normalization, the output $\mathbf{z}$ is fed into a linear layer, which maps it to the critic's value function predictions.

**Ensemble of Networks.** To further enhance stability and mitigate overestimation bias common in offline RL, FANS employs an ensemble of $M$ independently parameterized critic networks within the Actor-Critic framework. Each critic head $Q^{(m)}$, where $m = 1, 2, \ldots, M$, produces a separate estimate of the value function for a given state-action pair $(s, a)$. The ensemble output is computed as the average of these estimates:

$$Q_{\text{ensemble}}(s, a) = \frac{1}{M} \sum_{m=1}^{M} Q^{(m)}(s, a). \tag{4}$$

By aggregating multiple value predictions, this ensemble approach captures epistemic uncertainty and helps prevent the overestimation of OOD actions. During actual implementation, FANS operates within a standard Actor-Critic (AC) framework. The actor $\pi(a|s)$ is trained to maximize returns using value estimates from the critic ensemble.

**Critic Target.** Bootstrapped targets are computed using target networks $\pi_{\text{tgt}}$ and $Q_{\text{tgt}}^{(m)}$:

$$Q_{\text{target}}(s, a) = r(s, a) + \gamma \cdot \min_{m} Q_{\text{tgt}}^{(m)}(s', \pi_{\text{tgt}}(s')). \tag{5}$$

**Critic Loss.** Each critic minimizes MSE to the shared target. The ensemble critic loss is:

$$\mathcal{L}_{\text{critic}} = \sum_{m=1}^{M} \mathbb{E}_{(s,a)\sim\mathcal{D}} \left[ \left( Q^{(m)}(s, a) - Q_{\text{target}}(s, a) \right)^2 \right]. \tag{6}$$

**Actor Loss.** The actor is optimized to maximize the ensemble value:

$$\mathcal{L}_{\text{actor}} = -\mathbb{E}_{s\sim\mathcal{D}} \left[ Q_{\text{ensemble}}(s, \pi(s)) \right]. \tag{7}$$

## 4 Experiments

In this section, we conduct experiments to validate the effectiveness of FANS. First, we compare FANS with various state-of-the-art offline RL algorithms across multiple standard D4RL benchmarks [43] to evaluate its overall performance improvements. Then, to verify the motivation behind FANS − specifically, mitigating Q-value overestimation and enhancing generalization − we design targeted experiments that assess these properties in controlled settings. Finally, we perform ablation studies to analyze the contribution of each component in the framework and validate their necessity. The appendix includes experiments combining FANS with other baselines, evaluations on additional tasks, analyses of FANS's structural extensions, and learning curves omitted from the main text. Code is available at `https://github.com/DkING-lv6/FANS`.

### 4.1 Main Results

**Experimental Setup.** One of our main focuses is to demonstrate that FANS can yield substantial performance gains through minimal modifications to the network architecture. To this end, we adopt the simplest actor-critic framework, TD3, as our base algorithm. It is important to note that we do not incorporate any offline-specific constraints such as behavior cloning (BC), in order to purely isolate and showcase the effect of FANS. We evaluate the algorithm on MuJoCo locomotion tasks from the D4RL benchmark. For simply, we abbreviate the names of the datasets in all the tables as follows: {**halfcheetah** → **ha, hopper** → **ho, walker2d** → **wa, medium** → **m, medium-replay** →

Table 2: Performance comparison on D4RL locomotion tasks over the final ten evaluations and five seeds (normalized scores). We **bold** the highest mean.

| Tasks | TD3BC | AWAC | CQL | IQL | ReBRAC | SAC-N | EDAC | DT | **TD3 +FANS** |
|---|---|---|---|---|---|---|---|---|---|
| ha-m | 48.1 ±0.2 | 49.5 ±0.6 | 47.0 ±0.2 | 48.3 ±0.2 | 64.0 ±0.7 | **68.2** ±**1.3** | 67.7 ±1.0 | 42.2 ±0.3 | 66.6 ±0.8 |
| ha-mr | 44.8 ±0.6 | 44.7 ±0.7 | 45.0 ±0.3 | 44.5 ±0.2 | 51.2 ±0.3 | 60.7 ±1.0 | **62.1** ±**1.1** | 38.9 ±0.5 | 55.9 ±1.5 |
| ha-me | 90.8 ±6.0 | 93.6 ±0.4 | 95.6 ±0.4 | 94.7 ±0.5 | 103.8 ±3.0 | 99.0 ±9.3 | **104.8** ±**0.6** | 91.6 ±1.0 | 102.8 ±3.4 |
| ho-m | 60.4 ±3.5 | 74.5 ±9.1 | 59.1 ±3.8 | 67.5 ±3.8 | 102.3 ±0.2 | 40.8 ±9.9 | 101.7 ±0.3 | 65.1 ±1.6 | **104.6** ±**0.9** |
| ho-mr | 64.4 ±21.5 | 96.4 ±5.3 | 95.1 ±5.3 | 97.4 ±6.4 | 95.0 ±6.5 | 100.3 ±0.8 | 99.7 ±0.8 | 81.8 ±6.9 | **103.2** ±**1.1** |
| ho-me | 101.2 ±9.1 | 52.7 ±37.5 | 99.3 ±10.9 | 107.4 ±7.8 | 109.5 ±2.3 | 101.3 ±11.6 | 105.2 ±10.1 | 110.4 ±0.3 | **113.3** ±**1.4** |
| wa-m | 82.7 ±4.8 | 66.5 ±26.0 | 80.8 ±3.3 | 80.9 ±3.2 | 85.8 ±0.8 | 87.5 ±0.7 | 93.4 ±1.4 | 67.6 ±2.5 | **101.0** ±**1.6** |
| wa-mr | 85.6 ±4.0 | 82.2 ±1.1 | 73.1 ±13.2 | 82.2 ±3.0 | 84.2 ±2.3 | 79.0 ±0.5 | 87.1 ±2.8 | 59.9 ±2.7 | **98.3** ±**2.0** |
| wa-me | 110.0 ±0.4 | 49.4 ±38.2 | 109.6 ±0.4 | 111.7 ±0.9 | 111.9 ±0.4 | 114.9 ±0.4 | 114.8 ±0.7 | 107.1 ±1.0 | **118.1** ±**0.4** |
| Avg. | 76.5 | 67.7 | 78.3 | 81.6 | 89.7 | 83.5 | 92.9 | 73.8 | **96.0** |

**mr, medium-expert → me}**. For the baseline algorithms, we report results at 1M gradient steps, either by re-running the official implementations or directly adopting the values reported in their original papers. For our method, we conduct experiments using five random seeds and report the mean normalized score averaged over the final ten evaluations.

**Baselines.** We compare our method with several representative or state-of-the-art offline RL algorithms: (i) TD3+BC [3] combines TD3 with behavior cloning by adding a supervised loss to constrain policy updates toward the dataset actions; (ii) AWAC [44] accelerates offline learning by weighting advantages in actor-critic updates to prioritize high-value actions; (iii) CQL [8] introduces a conservative penalty on Q-values to prevent overestimation for unseen actions in offline settings; (iv) IQL [9] avoids explicit behavior cloning or importance sampling by selectively updating Q-values, V-values, and policy via expectile regression; (v) ReBRAC [45] enhances stability by regularizing the policy using behavior cloning and applying conservative Q-function updates; (vi) SAC-N extends SAC with an ensemble of Q-networks and a conservative penalty to improve performance and robustness; (vii) EDAC [12] improves value estimation in offline RL by decorrelating gradients across Q-networks in an ensemble; (viii) DT [46] reframes RL as a sequence modeling problem by training a transformer to predict actions conditioned on returns and past trajectories.

Table 2 provides a comprehensive comparison of our proposed FANS method against several state-of-the-art offline RL algorithms discussed above. Bolded values represent the highest normalized scores achieved for each task, while the ± denotes the standard deviation computed over five seeds. The results indicate that integrating the FANS framework into the structurally simple Actor-Critic algorithm TD3 yields the best average performance (Avg.) across all evaluated tasks. Notably, TD3+FANS demonstrates superior performance in most scenarios, with particularly pronounced improvements observed in the hopper and walker2d environments. Collectively, these results substantiate that employing a succinct yet effective network architecture can markedly enhance algorithmic performance, underscoring the significant potential and practical utility of minimalist architectural modifications in offline RL.

## 4.2 Validation of FANS in Mitigating Overestimation

In offline RL, overestimation is a common issue, particularly severe when encountering sparse or OOD state-action pairs caused by distributional shifts in the data. To address this challenge, our FANS framework incorporates several architectural designs. Residual blocks promote the learning of stable and low-frequency value functions through identity mapping pathways, helping to reduce overfitting and overestimation. The smooth nature of Gaussian activation functions guides the optimization towards flatter regions of the loss landscape, further reducing overestimation tendencies. Layer normalization stabilizes the training process by normalizing activations within each layer, mitigating risks of gradient explosion and vanishing, thereby improving value estimation accuracy. Finally, the ensemble method leverages multiple independent networks to collaboratively evaluate values, effectively suppressing overly optimistic estimates from individual models and enhancing the robustness and stability of value predictions.

To evaluate whether FANS mitigates overestimation, we measure value estimation accuracy by comparing each algorithm's Q-value predictions against the discounted Monte Carlo returns of its policy trajectories (ground truth). We quantify over- or underestimation by subtracting the ground truth from the predicted Q-value and dividing by the ground truth, where values greater or less than 0 indicate overestimation or underestimation. Averaged across six D4RL datasets (ha-m, ha-mr, ho-m, ho-mr, wa-m, wa-mr), FANS consistently achieves the most accurate Q-value estimation, with TD3BC also showing relatively precise predictions (Figure 2). This demonstrates that FANS

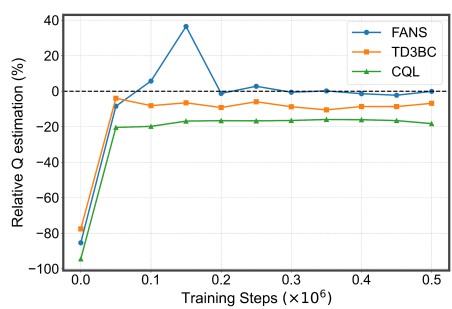

Figure 2: Relative Q estimation.

and TD3BC provide superior value calibration − avoiding the severe overestimation of vanilla TD3 and the strong underestimation of CQL. Notably, TD3BC's reasonable value accuracy does not translate to superior performance, indicating that accurate value estimation alone is insufficient.

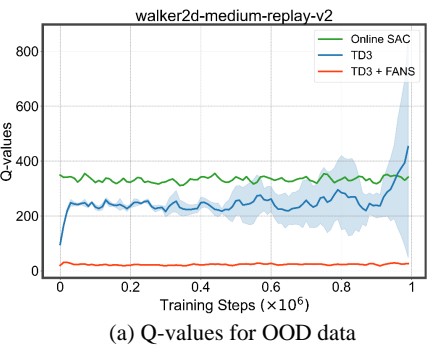

(a) Q-values for OOD data

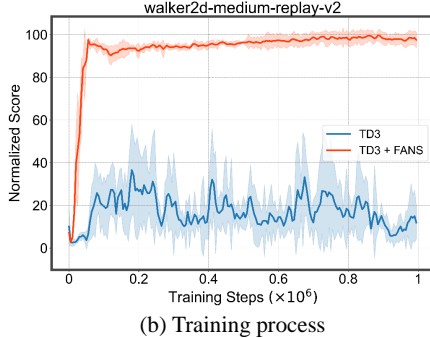

(b) Training process

Figure 3: Evaluation of overestimation mitigation: Q-value estimates on OOD data and corresponding performance of TD3 and TD3 + FANS.

Additionally, we conduct experiments comparing the Q-value estimations for OOD data. We randomly extract 100 groups (each containing 30 samples) from the walker2d-medium-replay dataset in the D4RL benchmark and use the remaining samples to train offline RL models. These extracted samples serve as OOD data relative to the current training set. We run Soft Actor-Critic (SAC) [47] on the MuJoCo walker2d-v2 environment and save the Q-value function model at 3M steps, then estimate these OOD data to obtain a standard Q-value (the curves of Online SAC in Figure 3(a)). Furthermore, we evaluate the TD3 algorithm and our proposed TD3+FANS on the dataset with OOD samples removed as described above. We periodically assess the Q-value estimates on the OOD data, with the average values reported in Figure 3(a). The results demonstrate that the baseline TD3 consistently produces elevated Q-value estimates for OOD inputs, ultimately leading to training collapse due to severe overestimation. In contrast, TD3+FANS maintains consistently lower Q-value estimates on OOD data, effectively mitigating overestimation issues. This advantage is further corroborated by the performance curves presented in Figure 3(b).

### 4.3 Revealing FANS's Fundamental Advantage: Generalization Control

#### 4.3.1 Theoretical Analysis using Neural Tangent Kernel (NTK)

To investigate why FANS outperforms TD3BC despite similar value accuracy, we analyze generalization patterns using Neural Tangent Kernel (NTK) [48]. When updating Q-values for $(s, a)$ using TD learning, the change at any $(\bar{s}, \bar{a})$ is governed by the NTK $k_\phi(\bar{s}, \bar{a}, s, a)$. A simple derivation in [48] is given in Appendix Section B. We track the normalized NTK improvement based on the NTK of initialized network (training step=0) and explore three OOD query types of $(\bar{s}, \bar{a})$ during training:

- $(s, \pi(s)), s \in D$
- $(s, \pi(s) + \epsilon), s \in D, \epsilon$ is noise
- $(s', a') \in D$ which is next state and next action (method of DR3 [49])

Due to the high dimensionality of $\nabla_\phi Q_\phi(s, a)$, the direct computation of $\underset{s,a\sim\mathcal{D}}{\mathbb{E}}|k_\phi(s, \pi(s), s, a)|$ is computationally prohibitive, a method is adopted by approximating $\Delta(\phi)$ with the contribution solely from the last layer parameters and obtain the $\underset{s,a\sim\mathcal{D}}{\mathbb{E}}|\Phi(s, \pi(s))^\top \Phi(s, a)|$, where $\Phi(s, a)$ signifies the representation of state-action pairs, which is the output of penultimate layer of Q-network.

We present the average results across six D4RL datasets (ha-m, ha-mr, ho-m, ho-mr, wa-m, wa-mr) logged every 100K step during 500K training steps (makes sure convergence) in Figure 4.

The significantly lower NTK values in FANS demonstrate a fundamental suppression of pathological generalization patterns. This occurs because the kernel $k_\phi(\bar{s}, \bar{a}, s, a)$ acts as a generalization amplifier: High values force TD-errors from ID data $(s, a)$ to distort OOD values $Q(\bar{s}, \bar{a})$. During offline RL, this propagates and amplifies extrapolation errors (especially dangerous for actions $\pi(s)$ near distribution boundaries). Our design directly counteracts this through: (1) residual connections that maintain stable feature baselines via identity mappings, preventing chaotic error propagation; (2) layer normalization that constrain feature co-adaptation by enforcing per-sample feature scale invariance and reducing sensitivity to

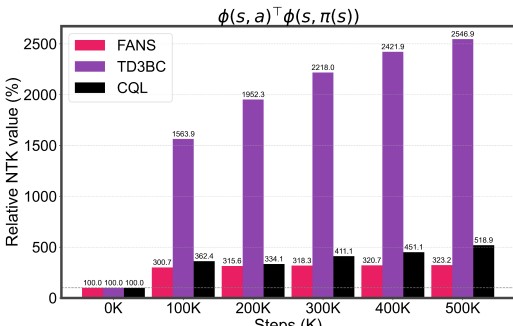

Figure 4: The NTK value of $(s, \pi(s)), s \in D$. Other two modes are presented in the Appendix.

aberrant activation patterns. These mechanisms collectively enhance the model's expressive capacity and mitigates harmful generalization errors, thus shows superior performance than baselines.

#### 4.3.2 Comparison with generalization-centric methods

The FANS framework promotes smoother loss surfaces during training, encouraging models to converge in flatter regions, which leads to better performance on unseen data. This structural design enhances the model's generalization ability, even when facing distribution shifts between the training and test data. To validate this, we first compare our method with several representative approaches designed to improve generalization: DOGE [28], TSRL [29], SPOT [5], POR [30], PRDC [7], STR [16], DIFFUION-QL [25], and CQL+ADS [18], as shown in Table 3. Among the many advanced methods associated with improving generalization, our approach remains highly competitive, achieving the best average performance.

#### 4.3.3 Validation under limited-data setting

Inspired by [29], we design small-sample experiments to test model performance with limited data to evaluate the generalization ablility of FANS. We reduce the training dataset to 5% of the full data and choose TD3 as the baseline algorithm. Note that FANS is implemented using a simple AC framework, without incorporating any advanced offline RL techniques such as behavior cloning (BC) constraints. The learning curves of different algorithms are shown in Figure 5.

Table 3: Performance comparison on D4RL locomotion tasks over the final ten evaluations and five seeds (normalized scores). We **bold** the highest mean.

| Tasks | DOGE | TSRL | SPOT | POR | PRDC | STR | DIFFUSION -QL | CQL +ADS | **TD3 +FANS** |
|-------|------|------|------|-----|------|-----|----------------|----------|---------------|
| ha-m  | 45.3 | 48.2 | 58.4 | 48.8 | 63.5 | 51.8 | 51.1 | **73.9** | 66.6 |
| ho-m  | 98.6 | 86.7 | 86.0 | 78.6 | 100.3 | 101.3 | 90.5 | 101.0 | **104.6** |
| wa-m  | 86.8 | 77.5 | 86.4 | 81.1 | 85.2 | 85.9 | 87.0 | 91.3 | **101.0** |
| ha-mr | 42.8 | 42.2 | 52.2 | 43.5 | 55.0 | 47.5 | 47.8 | 49.6 | **55.9** |
| ho-mr | 76.2 | 78.7 | 100.2 | 98.9 | 100.1 | 100.0 | 101.3 | 102.4 | **103.2** |
| wa-mr | 87.3 | 66.1 | 91.6 | 76.6 | 92.0 | 85.7 | 95.5 | 93.7 | **98.3** |
| ha-me | 78.7 | 92.0 | 86.9 | 94.7 | 94.5 | 94.9 | 96.8 | 93.5 | **102.8** |
| ho-me | 102.7 | 95.9 | 111.4 | 99.3 | 109.2 | 111.9 | 111.1 | 113.3 | 113.3 |
| wa-me | 110.4 | 109.8 | 112.0 | 109.1 | 111.2 | 110.2 | 110.1 | 112.1 | **118.1** |
| Avg.  | 81.0 | 77.5 | 85.9 | 80.1 | 90.1 | 87.7 | 87.9 | 92.3 | **96.0** |

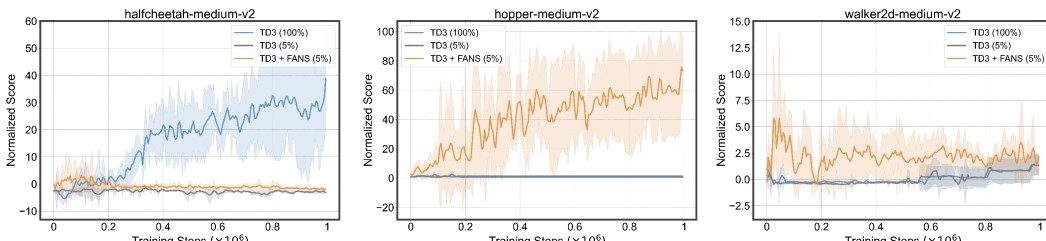

Figure 5: The performance of different algorithms under limited data. A ratio of 100% denotes training on the entire dataset, while 5% corresponds to using only one-twentieth of the full data.

Figure 5 clearly shows that when the training data is significantly reduced, FANS consistently demonstrates superior generalization ability across all tasks compared to the baseline algorithms. It is worth noting that, in the hopper and walker2d tasks, FANS trained on limited data even outperform the baseline algorithms trained on the full dataset. These results strongly indicate that FANS's structured design enables more effective feature extraction in data-scarce environments, thereby enhancing model robustness and generalization performance.

## 4.4   Ablation Study

Table 4 summarizes an ablation study evaluating the impact of each component in the proposed FANS architecture across multiple D4RL MuJoCo tasks. The results demonstrate that every module contributes to the overall performance, though to varying degrees.

Notably, the residual block and the ensemble mechanism emerge as the most critical components of the FANS framework. The removal of the residual block leads to the most substantial degradation in performance, underscoring its essential role in stabilizing learning and enhancing value approximation, likely by improving gradient flow and increasing network expressivity. Similarly, the ensemble mechanism contributes significantly to overall robustness; its absence results in a marked decline in performance, suggesting that aggregating multiple sub-policies effectively reduces variance. The final column in Table 4 includes the parameter $M$, representing the number of ensembles in FANS. The $M$ ranges from $[2, 3, 5]$, and we report the best-performing $M$ for each task. Different tasks exhibit varying needs for ensemble size.

The ablation of the Gaussian activation function also leads to a noticeable performance drop, as its core function of introducing stochasticity during policy learning helps enhance generalization and reduce the risk of overfitting to static offline datasets. In addition, layer normalization, while resulting in relatively smaller performance drops upon ablation, still provides meaningful benefits by stabilizing training dynamics and promoting smoother, more consistent learning behavior.

Table 4: Ablation study of FANS. $M$ is the number of ensemble in FANS.

| Tasks | TD3+FANS w/o Residual | TD3+FANS w/o Gaussian | TD3+FANS w/o LayNorm | TD3+FANS w/o Ensemble | TD3 + FANS | |
|---|---|---|---|---|---|---|
| | | | | | Score | $M$ |
| ha-m | $66.4 \pm 1.3$ | $66.2 \pm 0.8$ | $64.3 \pm 3.5$ | $66.6 \pm 0.8$ | $\mathbf{66.6 \pm 0.8}$ | 2 |
| ho-m | $21.0 \pm 19.3$ | $62.9 \pm 30.5$ | $99.6 \pm 6.9$ | $79.6 \pm 26.6$ | $\mathbf{104.6 \pm 0.9}$ | 5 |
| wa-m | $6.5 \pm 4.3$ | $7.7 \pm 7.8$ | $91.9 \pm 2.3$ | $6.7 \pm 0.4$ | $\mathbf{101.0 \pm 1.6}$ | 5 |
| ha-mr | $55.1 \pm 0.5$ | $54.4 \pm 1.2$ | $55.6 \pm 1.3$ | $55.1 \pm 1.3$ | $\mathbf{55.9 \pm 1.5}$ | 3 |
| ho-mr | $41.7 \pm 7.2$ | $95.1 \pm 4.5$ | $99.2 \pm 5.6$ | $44.6 \pm 6.3$ | $\mathbf{103.2 \pm 1.1}$ | 3 |
| wa-mr | $35.6 \pm 11.3$ | $80.2 \pm 22.1$ | $95.1 \pm 1.7$ | $75.4 \pm 23.5$ | $\mathbf{98.3 \pm 2.0}$ | 3 |
| ha-me | $64.5 \pm 21.1$ | $102.6 \pm 3.1$ | $51.1 \pm 4.7$ | $28.4 \pm 1.4$ | $\mathbf{102.8 \pm 3.4}$ | 5 |
| ho-me | $1.5 \pm 0.3$ | $30.3 \pm 8.2$ | $41.3 \pm 13.8$ | $1.6 \pm 0.8$ | $\mathbf{113.3 \pm 1.4}$ | 5 |
| wa-me | $-0.2 \pm 0.1$ | $89.1 \pm 21.0$ | $115.2 \pm 0.4$ | $16.5 \pm 10.7$ | $\mathbf{118.1 \pm 0.4}$ | 5 |
| Avg. | 32.3 | 65.4 | 79.3 | 41.6 | **96.0** | - |

In addition, our method considers residual only in the critic network. We provide ablation experiments to demonstrate that incorporating residual in the actor is unsuitable for offline RL. As the results presented in Table 5, the Control Setup refers to considering residual connections on both the actor and critic, which degrades the overall effectiveness. This degradation can be attributed to the residual connections in the actor causing the action output to be constrained within a limited range, effectively restricting the learning capacity within a narrow decision space. Consequently, this limitation hampers the actor's ability to learn optimal behaviors. Based on these observations, we adopt the design choice of applying residual connections exclusively in the critic network, where they facilitate stable value estimation without constraining the action space, thereby achieving better overall performance.

Table 5: Ablation study on residual placement: performance drops when residual blocks are added to the actor network.

| Structure | Control Setup | TD3 + FANS |
|---|---|---|
| Actor + Residual | ✓ | × |
| Critic + Residual | ✓ | ✓ |
| ha-m | $63.8 \pm 0.1$ | $\mathbf{66.6 \pm 0.8}$ |
| ho-m | $21.8 \pm 19.8$ | $\mathbf{104.6 \pm 0.9}$ |
| wa-m | $50.3 \pm 35.2$ | $\mathbf{101.0 \pm 1.6}$ |

## 5 Conclusion

In this work, we introduce FANS, a novel network architecture framework designed to tackle the unique generalization challenges of offline RL. By integrating residual blocks, Gaussian activation functions, layer normalization, and model ensembling, FANS systematically steers optimization toward flatter minima, thereby enhancing stability and reducing overfitting. Our comprehensive analyses reveal the individual and combined effects of these components in promoting smoother optimization landscapes and lowering variance. These results highlight the critical role of architectural design as a complementary and effective approach to advancing offline RL performance, opening promising avenues for future research. Future work will explore architectures better suited for offline-to-online settings, aiming to achieve more stable and safe policies that can rapidly adapt to the online environment.

## Acknowledgments and Disclosure of Funding

The work is supported by the National Natural Science Foundation of China (No. U21A20473, No. 62276160), the Fundamental Research Program of Shanxi Province (No. 202503021212091, No. 202403021222153), the Shanxi Provincial Natural Science Foundation General Project (No. 202203021211294), the Shanxi Provincial Overseas Study Fund Project (No. 20240002), the Scientific and Technological Innovation Programs of Higher Education Institutions in Shanxi (No. 2025L001).

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

# A Implementation details

All experiments run on a server equipped with an Intel® Xeon® Gold 6254 CPU @ 3.10GHz and NVIDIA GeForce RTX 3090 GPU.

Table 6: Detailed hyperparameter settings for the proposed FANS framework.

| | Hyperparameter | Value / Range | Description |
|---|---|---|---|
| **Architecture** | Network depth | 2 | Number of residual blocks in the critic. |
| | Hidden dimension ($d$) | 256 | Width of hidden layers in both actor and critic. |
| | Activation (actor) | ReLU | Activation function used in the actor network. |
| | Activation (critic) | Gaussian | Smooth, bounded nonlinearity encouraging flatter optimization landscapes. |
| | Residual design | Pre-activation | Structure: LayerNorm $\rightarrow$ Linear $\rightarrow$ Gaussian $\rightarrow$ Linear (with skip connection). |
| | Output normalization | LayerNorm | Applied after the final residual block to stabilize representation magnitudes. |
| **Training** | Optimizer | Adam | Optimizer used for both actor and critic. |
| | Learning rate | $1 \times 10^{-4}$ | Learning rate for both actor and critic. |
| | Batch size | 256 | Samples used per optimization step. |
| | Discount factor ($\gamma$) | 0.99 | Reward discount coefficient. |
| | Training steps | 1M | Total number of gradient update iterations. |
| | Seeds | {0, 10, 100, 1000, 10000} | Five seeds used for all reported averages. |
| | Ensemble size ($M$) | {2, 3, 5} | Number of independently parameterized critic networks; best $M$ per task is reported. |

# B Details of Neural Tangent Kernel (NTK)

**Parameter update** for $(s, a)$:

$$\phi' = \phi + (\mathcal{T}Q_\phi(s, a) - Q_\phi(s, a)) \nabla_\phi Q_\phi(s, a) \tag{8}$$

where $\mathcal{T}$ is the target operator (learning rate omitted).

**Q-value change** at $(\bar{s}, \bar{a})$ by Taylor expansion at the pre-update parameter $\phi$:

$$Q_{\phi'}(\bar{s}, \bar{a}) = Q_\phi(\bar{s}, \bar{a}) + \nabla_\phi Q_\phi(\bar{s}, \bar{a})^\top (\phi' - \phi) + \mathcal{O}(\|\Delta\phi\|^2) \tag{9}$$

**Generalization via NTK by plugging the second equation into the first one**:

$$Q_{\phi'}(\bar{s}, \bar{a}) = Q_\phi(\bar{s}, \bar{a}) + \left[k_\phi(\bar{s}, \bar{a}, s, a)\right] (\mathcal{T}Q_\phi(s, a) - Q_\phi(s, a)) + \mathcal{O}(\|\Delta\phi\|^2) \tag{10}$$

where

$$k_\phi(\bar{s}, \bar{a}, s, a) = \nabla_\phi Q_\phi(\bar{s}, \bar{a})^\top \nabla_\phi Q_\phi(s, a) \tag{11}$$

**NTK interpretation**:

- **High $k_\phi$** $\Rightarrow$ Prominent generalization: $Q(\bar{s}, \bar{a})$ changes significantly in sync with the TD error updating on $Q(s, a)$.
- **Low $k_\phi$** $\Rightarrow$ Minimal generalization (tabular-like behavior when $k_\phi = 0$).

**Key Insight:** The NTK $k_\phi(\bar{s}, \bar{a}, s, a)$ quantifies how much updating $Q(s, a)$ affects $Q(\bar{s}, \bar{a})$, acting as a generalization metric for TD learning.

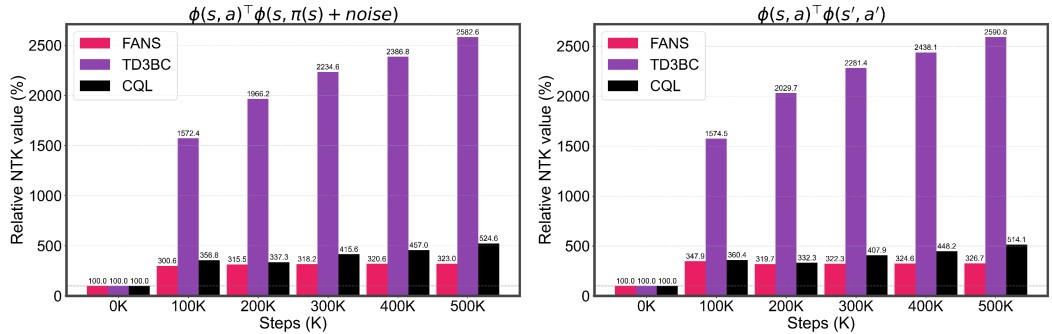

Figure 6: The NTK values of $(s, \pi(s) + noise)$ and $(s', a')$.

## C Additional Results

### C.1 Additional Evaluations

In addition to the integration of FANS with TD3 presented in the main text, we also applied FANS to AWAC. As shown in Table 7, AWAC + FANS demonstrates a significant performance improvement over the original AWAC algorithm across a range of offline RL tasks. Notably, in challenging tasks such as *hopper-medium-expert* and *walker2d-medium-expert*, AWAC + FANS boosts the scores from 52.7 to 110.3 and from 49.4 to 109.6, respectively. These improvements are not only substantial but also come with significantly reduced variance, indicating more stable and reliable policy behavior.

Moreover, the average performance across all tasks increases from 67.7 to 83.9, further highlighting the general and consistent enhancement brought by the FANS module. These results validate the effectiveness of our approach across diverse environments and demonstrate its potential as a general-purpose enhancement to existing offline RL methods.

Table 7: Performance comparison on D4RL locomotion tasks over the final ten evaluations and five seeds (normalized scores). We **bold** the highest mean.

| Tasks | AWAC | **AWAC + FANS** |
|---|---|---|
| halfcheetah-medium | **49.5 ± 0.6** | 48.9 ± 0.5 |
| halfcheetah-medium-replay | 44.7 ± 0.7 | **44.9 ± 0.2** |
| halfcheetah-medium-expert | 93.6 ± 0.4 | **94.4 ± 0.4** |
| hopper-medium | 74.5 ± 9.1 | **76.2 ± 6.1** |
| hopper-medium-replay | 96.4 ± 5.3 | **99.8 ± 2.0** |
| hopper-medium-expert | 52.7 ± 37.5 | **110.3 ± 0.7** |
| walker2d-medium | 66.5 ± 26.0 | **81.9 ± 0.7** |
| walker2d-medium-replay | 82.2 ± 1.1 | **88.9 ± 3.8** |
| walker2d-medium-expert | 49.4 ± 38.2 | **109.6 ± 1.2** |
| Average | 67.7 | **83.9** |

## C.2 Analyses of FASN's Structural Extensions

We conduct a systematic scaling analysis of the critic network by varying its depth (1–4) and width (64–512). For width scaling, the critic depth is fixed at 2 blocks; for depth scaling, the critic width is set to 256, following our default setup.

The results show that the best performance is achieved when the critic has depth = 2 and width = 256, yielding high returns with low standard deviations across all three tasks (*halfcheetah-medium*, *hopper-medium*, and *walker2d-medium*), indicating strong stability.

Overall, increasing the depth and width of the critic generally leads to performance improvements, suggesting that higher model capacity enhances the representational power of the value estimator.

However, we also observe significant instability under certain configurations (e.g., depth = 4 or width = 64/512), particularly on the *hopper-medium* and *walker2d-medium* tasks, where the standard deviations are notably large. This highlights the trade-off between model capacity and training stability, and the importance of balancing expressiveness with generalization.

In summary, properly scaling the critic network can significantly boost performance, but must be done with care to avoid instability.

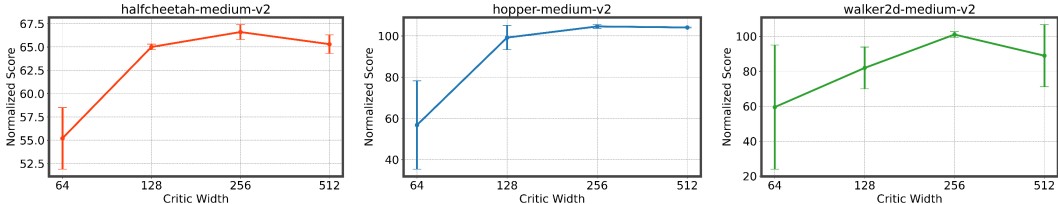

Figure 7: Performance of TD3 with FANS by varying width for the critic network.

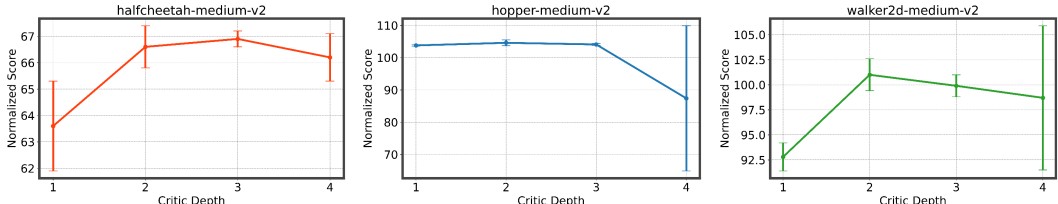

Figure 8: Performance of TD3 with FANS by varying depth for the critic network.

## C.3 Learning Curves

The learning curves of TD3 + FANS for all tasks corresponding to Table 2 in the main text are shown in the figure below.

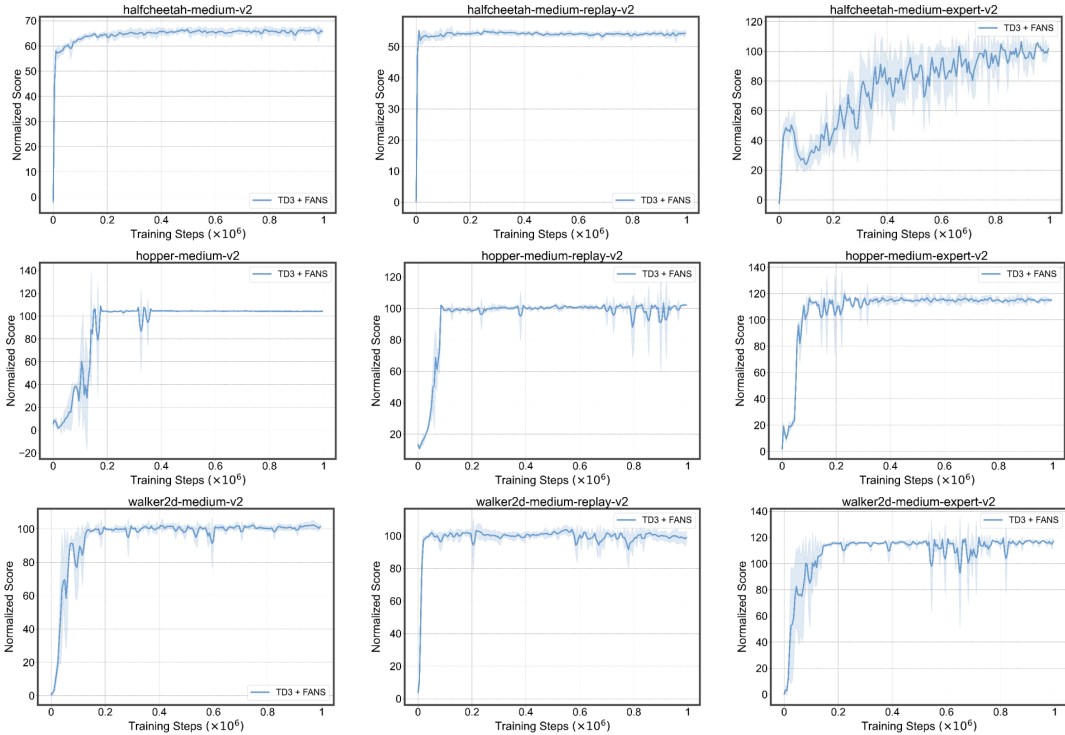

Figure 9: Performance of TD3 with FANS on different mujoco tasks.

