# OpenReview forum: "FANS: A Flatness-Aware Network Structure for Generalization in Offline Reinforcement Learning"
_NeurIPS.cc/2025/Conference — NeurIPS 2025 poster_

### Official Review · Reviewer_6MWm · 2025-06-26

**Clarity:** 2
**Significance:** 1
**Originality:** 1
**Rating:** 3
**Confidence:** 4

**Summary:**

This paper introduces FANS, a novel method for mitigating the Q-value overestimation problem in offline reinforcement learning through minimal network modifications. FANS integrates Residual connections, Gaussian Activations, Layer Normalization, and a model Ensemble. FANS combined with TD3 is shown to outperform other offline RL methods. Furthermore, the paper presents a series of ablation studies to validate the effectiveness of the FANS components.

**Questions:**

- Is there a specific reason why the four components of FANS are most effective when used together? It would be more convincing if you could provide an ablation study that demonstrates the synergy between these components, rather than just their cumulative benefit.
- The paper's claim of reducing Q-value overestimation is not clearly substantiated against other offline algorithms. Could you please provide a direct comparison of the Q-values against other state-of-the-art offline RL methods to better illustrate the improvements?
- To demonstrate the generalizability of your method, could you provide performance comparisons against other offline algorithms on the D4RL **AntMaze** and **Adroit** benchmark suites?
- The FANS architecture does not seem to be inherently limited to the offline RL setting. Have you considered if this architecture could also improve performance in other reinforcement learning paradigms where distribution shift is a key problem, such as offline-to-online RL, continual RL, or multitask RL?

**Ethical Concerns:**

["NO or VERY MINOR ethics concerns only"]

**Final Justification:**

**Final Justification for Score (Score: 3)**

After considering the authors’ rebuttal and reflecting on the discussion, I provide the following justification for my score.

**Resolved Issues:**
- The authors clarified the motivation behind comparing TD3 and TD3+FANS in Figures 2 and 3. I now have a better understanding that these experiments aim to demonstrate the effectiveness of the proposed structured enhancement.
- Through additional baselines and experiments in diverse environments, the authors have shown that FANS performs well not only in MuJoCo but also in general offline RL settings, outperforming other algorithms.

**Unresolved Issues:**
- In the overestimation experiment (Figure 2), while the authors demonstrate that FANS reduces Q-value overestimation compared to TD3, they do not sufficiently address whether other offline RL methods would show similar improvements. It remains unclear why FANS should perform better than other offline RL algorithms specifically in mitigating Q-value overestimation.
- The authors correctly state that simply lowering Q-values for OOD data does not guarantee better performance. However, they do not provide a clear conceptual or quantitative criterion for what constitutes “appropriate” Q-value suppression, nor do they convincingly demonstrate that FANS satisfies such a criterion.

**Summary for Authors:**
- The rebuttal helped clarify the motivation and design of FANS.
- However, key concerns about baseline comparisons and the analysis of Q-value overestimation remain only partially addressed.
- Given the partial resolution, I have increased my score to 3, reflecting a more favorable view—but I still have some remaining concerns.

**Limitations:**

The authors addressed all limitations.

**Quality:**

2

**Strengths And Weaknesses:**

### **Strengths**

- **Clarity and Presentation:** The paper is exceptionally well-written, and the proposed method is presented with great clarity. This makes the core ideas and contributions easy for the reader to understand and follow.
- **Simplicity and Effectiveness:** A key strength of this work is its simplicity. The FANS method achieves significant performance improvements in offline reinforcement learning by introducing what appear to be minimal, yet effective, modifications to the network architecture.

### **Weaknesses**

- **Insufficient Ablation and Analysis of Components:** The paper lacks a rigorous experimental analysis of the FANS components (Residual connection, Gaussian Activation, Layer Normalization, and Model Ensemble). The current experiments do not adequately explain *why this specific combination* is critical or how the components synergize to reduce overestimation. For instance, there is no analysis showing that this combination is superior to other possible combinations or that it maximally reduces overestimation. As it stands, the performance gains could be interpreted as merely the cumulative benefit of four independent techniques, rather than a cohesive, synergistic solution.
- **Limited Scope of Evaluation Environments:** The empirical evaluation is conducted on a narrow set of environments. To robustly demonstrate the generalizability of FANS, it is crucial to include comparisons with other algorithms on more diverse and challenging offline RL benchmarks, such as the D4RL **AntMaze** and **Adroit** suites. It is common for algorithms to perform well on MuJoCo locomotion tasks but struggle with AntMaze or Adroit, so evaluation on these benchmarks is critical to validate the method's effectiveness.
- **Unclear Justification in Overestimation Analysis (Figure 2):** The intent of the Q-value overestimation experiment shown in Figure 2 is difficult to interpret. While the goal seems to be showing that TD3+FANS reduces Q-function overestimation, comparing it to online RL algorithms like the standard TD3 and SAC is not a meaningful benchmark. The dynamics of Q-value learning are fundamentally different in online versus offline settings. To be convincing, the analysis should demonstrate the advantages of FANS relative to other leading *offline* RL algorithms.
- **Inappropriate Baseline in Limited-Data Experiments (Figure 3):** The validation under a limited-data setting (Figure 3) suffers from a similar issue. Comparing an offline method to the online TD3 algorithm is not a meaningful evaluation of data efficiency. An online algorithm's performance is not a relevant baseline for an offline method. The experiment should instead demonstrate that FANS maintains a more stable and effective performance compared to other TD3-based *offline* algorithms when trained on a small fraction (e.g., 5%) of the available data.

---

> ### Author Rebuttal · Authors · 2025-07-31
>
> Thank you for your insightful comments. To comprehensively address your concerns, we will explain the following aspects in order: **inappropriate baselines**, **methods to reduce Q-value overestimation**, **ablation studies**, **discussions on other paradigms**, and **additional experiments**. The order may differ from your questions, but we aim to thoroughly resolve your main concerns. We sincerely appreciate the time and effort you have invested during the rebuttal process.
>
> **> 1. Response to Inappropriate Baseline:**
>
> As pointed out in the paper *Stabilizing Off-Policy Q-Learning via Bootstrapping Error Reduction*, online algorithms such as TD3 and SAC often **struggle to perform effectively in offline settings**. This has led to the development of offline variants like TD3+BC and SAC-N. However, these methods are **fundamentally built upon online algorithms**, enhanced through additional regularization or structural modifications. Similarly, our proposed **FANS framework is a structured enhancement tailored for offline RL**. TD3+FANS and TD3+BC can thus be viewed as different extensions of the same base algorithm.
>
> It is important to note that applying FANS on top of existing offline algorithms like TD3+BC would introduce **conceptual conflicts**. Specifically, BC regularization **enforces conservative behavior** by constraining the learned policy to stay close to the behavior policy, whereas FANS aims to **improve generalization beyond the behavior policy**. These objectives are inherently at odds. Therefore, we chose to apply FANS directly to the base online algorithm TD3 to **better isolate and evaluate the contribution of our structured framework**, without interference from other offline-specific techniques.
>
> That said, to further address your concern, we have **added experiments using TD3+BC as a baseline**. The below results support our analysis and demonstrate the validity of our design choice.
>
> | |ha-m|ho-m|wa-m|ha-mr|ho-mr|wa-mr|ha-me|ho-me|wa-me|
> |-|-|-|-|-|-|-|-|-|-|
> | TD3+BC+FANS|64.4±2.8|63.5±43.1|13.0±2.4|54.1±1.3|73.6±25.9| 58.1±33.3|68.3±7.3|18.4±7.0|13.6±5.0|
> | **TD3+FANS** |**66.6 ± 0.8**|**104.6±0.9**|**101.0±1.6**|**55.9±1.5**|**103.2±1.1**|**98.3±2.0**|**102.8±3.4**|**113.3±1.4**|**118.1±0.4**|
>
> Moreover, we would like to highlight that our **simple combination of TD3 and FANS achieves new SOTA performance** compared to a wide range of baselines, including recent SOTA methods and those focusing on generalization. Following the suggestion from Reviewer RRU4 (#R1), we have also added comparisons with several of these latest methods. Please refer to our **response to #R1, point 3**, for more details.
>
> **Regarding the concern about the "inappropriate baseline" in Figure 3**. The primary purpose of this experiment is to provide an intuitive illustration of how FANS can significantly improve the generalization ability of a base algorithm (e.g., TD3) under limited data conditions. **Figure 3 shows that while TD3 almost completely fails in this setting, FANS still achieves stable and effective performance, highlighting the potential of our structured enhancement in low-data regimes**. We agree that comparing FANS with other TD3-based offline algorithms under limited data could provide a more direct evaluation of data efficiency. While we emphasize that this experiment is not intended as a comprehensive SOTA comparison, we will consider adding such comparisons in future versions to further demonstrate FANS’s effectiveness in small-data scenarios. That said, **the focus of this experiment is to serve as a qualitative analysis that emphasizes the stability and generalization benefits FANS brings to the base policy**. In other parts of the paper, we have already conducted systematic comparisons between FANS and various mainstream generalization methods.
>
> **> 2. Response to Reduce Q-value Overestimation:**
>
> The purpose of Figure 2 is to demonstrate the effect of the FANS framework in reducing Q-value overestimation. One common misconception is to assume that lower Q-values for OOD data are always better. In reality, what matters is the relative magnitude of Q-values $-$ i.e., whether high Q-values for OOD data are properly suppressed. **Therefore, we should focus on whether the Q-value curve for OOD data exhibits signs of Q-value explosion**, which is a key indicator of overestimation. In Figure 2, it is evident that the online TD3 algorithm suffers from a Q-value explosion for OOD data, while introducing FANS to TD3 effectively prevents this issue. This highlights FANS’s ability to stabilize value estimation under distribution shift.
>
> In addition, **directly comparing the absolute scale of Q-values is not always meaningful**, as the values can differ by orders of magnitude due to implementation-specific factors. In FANS, the use of residual connections helps retain the original input’s scale, and the second linear layer adds new feature representations, which tends to maintain or even amplify the output magnitude. Moreover, LayerNorm normalizes each sample to have zero mean and unit variance within a feature dimension. While it does not change the overall expressive capacity, it can compress intermediate feature scales, thereby indirectly affecting the final Q-value range.
>
> **Another common misconception is that lower Q-values for OOD data necessarily lead to better performance**. As shown in the MCQ paper, overly conservative handling of OOD data can suppress potential generalization, leading to suboptimal outcomes. Thus, appropriately controlling Q-values for OOD data may be more promising, although the optimal strategy remains an open question.
>
> In summary, **the goal of our analysis is to show that FANS effectively reduces harmful Q-value overestimation for OOD samples**. For this reason, we directly compare TD3 and TD3+FANS in Figure 2. The online SAC curve, trained for 3 million steps, serves as a proxy for the "true" Q-value on OOD data. In contrast, directly comparing Q-values with other offline SOTA methods is not a suitable or reliable benchmark at this stage. **This experiment is also designed with reference to Sec. 5.2 "Value Estimation Error" from the paper *Improving Generalization in Offline Reinforcement Learning via Adversarial Data Splitting*.**
>
> **> 3. Response to Ablation Studies:**
>
> Our FANS framework is inspired by the well-established structured approach SimBa from the online setting. The structural differences between FANS and SimBa are as follows:
>
> |Comments|SimBa|FANS|
> |-|-|-|
> |Input Normalization Method|Running Statistics Normalization (RSNorm)|Dataset Mean Normalization|
> |Residual Feedforward Blocks|✔|✔|
> |Activation functions|ReLU|Gaussian|
> |Post-layer Normalization|✔|✔|
> |Ensemble|✘|✔|
>
> Compared to SimBa, FANS is specifically designed to enhance generalization in offline settings by additionally incorporating Gaussian activation functions and ensemble techniques. Therefore, on one hand, given that SimBa already provides a mature structured design for RL, there is no need to reiterate the analysis of the combination and effects of Residual Blocks and Layer Normalization.
>
> We further supplement our work by presenting results of directly applying SimBa in the offline setting, i.e., TD3+SimBa, as shown in the table below:
>
> ||ha-m|ho-m|wa-m|ha-mr|ho-mr|wa-mr|ha-me|ho-me|wa-me|
> |-|-|-|-|-|-|-|-|-|-|
> |TD3+SimBa|64.4±2.8|63.5±43.1|13.0±2.4|54.1±1.3|73.6±25.9|58.1±33.3|68.3±7.3|18.4±7.0|13.6±5.0|
> |**TD3+FANS**|**66.6±0.8**|**104.6±0.9**|**101.0±1.6**|**55.9±1.5**|**103.2±1.1**|**98.3±2.0**|**102.8±3.4**|**113.3±1.4**|**118.1±0.4**|
>
> The additional components introduced on top of SimBa $-$ the Gaussian activation function and ensemble techniques $-$ are well demonstrated in the ablation experiments in Table 4 of this paper. For instance, in the ho-me task, while the full FANS achieves 113.3, removing any single component causes severe performance drops (best ablated score: 41.3 without LayerNorm), revealing **nonlinear synergies between components**. Moreover, different tasks rely on distinct components (e.g., Gaussian noise is crucial for ho-m but LayerNorm dominates wa-m), proving **our components dynamically collaborate rather than providing fixed incremental gains**. This emergent synergy explains why the complete FANS consistently outperforms all partial implementations across diverse tasks.
>
> Moreover, while SimBa applies residual blocks to both the actor and the critic, our FANS framework further analyzes the effects of applying residual blocks separately to the actor and the critic in offline environments, as detailed in Table 5 of this paper.
>
> **> 4. Response to Discussions on other Paradigms:**
>
> This work focuses on offline RL and does not specifically discuss other RL paradigms. However, we believe that some of the design principles embodied by FANS $-$ such as guiding the optimization process toward flatter regions through structured design and reducing estimation variance to enhance generalization $-$ may offer valuable insights for other paradigms that are also heavily affected by distribution shifts. In offline-to-online RL, continual RL, or multitask RL, a core challenge is improving the policy’s adaptability to dynamically changing environments or tasks. The modular structure of FANS provides a potential architectural framework for further exploration of these challenges.
>
> It is important to emphasize that applying FANS to these paradigms would require appropriate extensions or adaptations tailored to their specific problem characteristics, such as incorporating task-specific memory mechanisms, adaptive modules, or dynamic sharing structures, rather than directly applying the existing architecture. Therefore, systematic investigations of FANS in other paradigms are left for future work.
>
> **> 5. Response to Additional Experiments:**
>
> We have supplemented the results on AntMaze and Adroit; please refer to our **response to Reviewer RRU4 (#R1), point 4**.

---

> ### Author Response · Authors · 2025-08-07
> **Further Response to Reviewer 6MWm (PART 1)**
>
> Dear Reviewer 6MWm:
>
> We sincerely appreciate your insightful feedback and recognition of FANS's contribution potential (raised score to 3). We now address the key concerns you mentioned with **theoretical analysis and empirical evidence**:
>
> ### **1. Resolving Ambiguity: FANS vs. Offline Baselines in Q-value Control**
>
> **New Experiment:** We quantitatively measure **value estimation accuracy** by comparing the discounted Monte Carlo return of each algorithm’s policy trajectories (ground truth) versus its Q-value predictions. To facilitate comparison, we quantified the level of Q-value over/underestimation for each algorithm by subtracting the ground truth from the predicted Q-value and dividing the result by the ground truth.  Greater/less than 0 means overestimation/underestimation. Average results across six D4RL datasets (walker2d-medium, walker2d-medium-replay, hopper-medium, hopper-medium-replay, halfcheetah-medium, halfcheetah-medium-replay) in the following table show:
>
> | Q esitimation | 0K | 50K | 100K | 150K | 200K | 250K | 300K | 350K | 400K | 450K | 500K |  |
> | --- | --- | --- | --- | --- | --- | --- | --- | --- | --- | --- | --- | --- |
> |  CQL    |  -94.45%  |  -20.40%  |  -19.82%  |  -16.85%  |  -16.59%  |  -16.71%  |  -16.50%  |  -15.94%  |  -16.07%  |  -16.56%  |  -18.26%  |  |
> |  TD3BC  |  -77.59%  |  -3.98%   |  -8.15%   |  -6.55%   |  -9.19%   |  -5.96%   |  -8.75%   |  -10.54%  |  -8.64%   |  -8.67%   |  -6.83%   |  |
> |  FANS   |  -85.37%  |  -8.45%   |  5.71%    |  36.47%   |  -1.25%   |  2.77%   |  -0.58%   |  0.17%   |  -1.42%   |  -2.26%   |  -0.11%   |  |
>
> As the training goes, FANS achieves the most accurate Q-value estimation. TD3BC also shows relatively accurate estimation. This proves FANS and TD3BC achieves **superior value calibration** – neither severe overestimating like vanilla TD3 nor severe underestimating like CQL. Crucially, while TD3BC also shows reasonable accuracy, its **inferior empirical performance** indicates that *value accuracy alone is insufficient*.
>
> ---
>
> ### **2. Revealing FANS's Fundamental Advantage:  Generalization Control**
>
> To investigate why FANS outperforms TD3BC despite similar value accuracy, we analyze **generalization patterns** using **Neural Tangent Kernel (NTK) [1]**. When updating $Q$-values for $(s,a)$ using TD learning, the change at any $(\bar{s},\bar{a})$ is governed by the **NTK** $k_\phi(\bar{s},\bar{a},s,a)$. A simple derivation in [1] is given:
>
> 1. **Parameter update** for $(s,a)$:
>
>     $$\phi' = \phi + \left( \mathcal{T} Q_\phi(s, a) - Q_\phi(s, a) \right) \nabla_\phi Q_\phi(s, a)$$
>
>     ($\mathcal{T}$: target operator, learning rate omitted)
>
> 2. **Q-value change** at $(\bar{s},\bar{a})$ by Taylor expansion at the pre-update parameter $\phi$:
>
>     $$Q_{\phi'}(\bar{s}, \bar{a}) = Q_\phi(\bar{s}, \bar{a}) + \nabla_\phi Q_\phi(\bar{s}, \bar{a})^\top (\phi' - \phi) + \mathcal{O}(\|\Delta\phi\|^2)$$
>
> 3. **Generalization via NTK by plugging the second Eq to the first one**:
>
>     $$Q_{\phi'}(\bar{s}, \bar{a}) = Q_\phi(\bar{s}, \bar{a}) +[ k_\phi(\bar{s},\bar{a},s,a) ] \left( \mathcal{T} Q_\phi(s, a) - Q_\phi(s, a) \right) + \mathcal{O}(\|\Delta\phi\|^2)$$
>
>     where $k_\phi(\bar{s},\bar{a},s,a)$ is $\nabla_\phi Q_\phi(\bar{s}, \bar{a})^\top \nabla_\phi Q_\phi(s, a)$
>
> 4. **NTK interpretation**:
>     - **High $k_\phi$** → Prominent generalization: $Q(\bar{s},\bar{a})$ changes significantly in sync with TD error updating on $Q(s,a)$
>     - **Low $k_\phi$** → Minimal generalization (tabular-like behavior when $k_\phi=0$)
>
> **Key Insight: The NTK $k_\phi(\bar{s},\bar{a},s,a)$ quantifies how much updating $Q(s,a)$ affects $Q(\bar{s},\bar{a})$*, acting as a generalization metric for TD learning.**
>
> ---
>
> **Key Experiment:** We track the normalized NTK improvement based on the NTK of initialized network (training_step=0) and explore three OOD query types of $(\bar{s}, \bar{a})$ during training:
>
> - $(s,\pi(s)), s \in D$;
> - $(s,\pi(s)+\epsilon),s \in D, \epsilon$ is noise;
> - $(s’,a’) \in D$ which is next state and next action (method of DR3[2])
>
> Due to the high dimensionality of $\nabla_\phi Q_\phi(s,a)$, the direct computation of $\underset{s, a \sim \mathcal{D}}{\mathbb{E}} |k_\phi(s, \pi(s), s, a)|$  is computationally prohibitive, a method is adopted by approximating $\Delta(\phi)$ with the contribution solely from the last layer parameters and obtain the $$\underset{s, a \sim \mathcal{D}}{\mathbb{E}} |\Phi(s, \pi(s))^{\top} \Phi(s, a)|$$
> where $\Phi(s, a)$ signifies the representation of state-action pairs, which is the output of penultimate layer of Q-network.

---

> ### Author Response · Authors · 2025-08-07
> **Further Response to Reviewer 6MWm (PART 2)**
>
> **Results**:
>
> We present the average results across six D4RL datasets (walker2d-medium, walker2d-medium-replay, hopper-medium, hopper-medium-replay, halfcheetah-medium, halfcheetah-medium-replay) logged every 50k step during 500K training steps (makes sure convergence)  in the following tables.
>
> | $\phi(s,a)*\phi(s,\pi(s))$ | 0K | 50K | 100K | 150K | 200K | 250K | 300K | 350K | 400K | 450K | 500K |
> | --- | --- | --- | --- | --- | --- | --- | --- | --- | --- | --- | --- |
> |  FANS  |  100.00%  |  256.15%  |  300.71%  |  311.70%  |  315.57%  |  318.95%  |  318.28%  |  320.06%  |  320.73%  |  318.37%  |  323.24%  |
> |  TD3BC  |  100.00%  |  1102.53%  |  1563.93%  |  1823.55%  |  1953.21%  |  2057.09%  |  2217.98%  |  2369.91%  |  2421.85%  |  2417.71%  |  2546.86%  |
> |  CQL  |  100.00%  |  397.76%  |  362.43%  |  332.08%  |  334.07%  |  356.53%  |  411.03%  |  443.30%  |  451.14%  |  492.92%  |  518.89%  |
>
> | $\phi(s,a)*\phi(s,\pi(s)+noise)$ | 0K | 50K | 100K | 150K | 200K | 250K | 300K | 350K | 400K | 450K | 500K |
> | --- | --- | --- | --- | --- | --- | --- | --- | --- | --- | --- | --- |
> |  FANS    |  100.00%  |  256.05%   |  300.58%    |  311.62%    |  315.52%    |  318.75%    |  318.13%    |  319.81%    |  320.59%    |  318.17%    |  322.95%    |
> |  TD3BC   |  100.00%  |  1107.28%  |  1572.41%   |  1833.99%   |  1966.19%   |  2071.72%   |  2234.95%   |  2386.81%   |  2438.27%   |  2433.68%   |  2565.00%   |
> |  CQL     |  100.00%  |  401.45%   |  365.81%    |  335.55%    |  337.33%    |  359.92%    |  415.60%    |  447.80%    |  455.02%    |  498.09%    |  524.57%    |
>
> | $\phi(s,a)*\phi(s',a')$ | 0K | 50K | 100K | 150K | 200K | 250K | 300K | 350K | 400K | 450K | 500K |
> | --- | --- | --- | --- | --- | --- | --- | --- | --- | --- | --- | --- |
> |  FANS    |  100.00%  |  259.12%   |  304.87%    |  316.03%    |  319.70%    |  323.04%    |  322.23%    |  323.82%    |  324.16%    |  321.62%    |  326.66%    |
> |  TD3BC   |  100.00%  |  1109.12%  |  1574.53%   |  1840.20%   |  1956.68%   |  2060.04%   |  2231.30%   |  2386.96%   |  2437.24%   |  2427.73%   |  2554.94%   |
> |  CQL     |  100.00%  |  396.22%   |  360.37%    |  330.02%    |  332.27%    |  354.53%    |  407.88%    |  441.90%    |  448.18%    |  488.07%    |  514.06%    |
>
> **Interpretation**:
>
> - The significantly lower NTK values in FANS demonstrate a **fundamental suppression of pathological generalization patterns**. This occurs because the kernel $k_\phi(\bar{s},\bar{a},s,a)$ acts as a *generalization amplifier:* High values force TD-errors from ID data $(s,a)$ to distort OOD values $Q(\bar{s},\bar{a})$. During offline RL, this propagates and amplifies extrapolation errors (especially dangerous for actions $\pi(s)$ near distribution boundaries).
> - Our design directly counteracts this through: (1) residual connections that maintain stable feature baselines via identity mappings, preventing chaotic error propagation; (2) layer normalization that constrain feature co-adaptation by enforcing per-sample feature scale invariance and reducing sensitivity to aberrant activation patterns. **These mechanisms collectively enhance the model’s expressive capacity and mitigates harmful generalization errors, thus shows superior performance than baselines.**
>
> ---
>
> ### **3. Summary**
>
> We agree a universal criterion for "optimal" OOD suppression remains open. However, our work reveals a **paradigm shift**: Offline RL's extrapolation error may stem more from **MLPs' poor approximation properties** than algorithmic limitations. FANS addresses via **two measurable advantages**:
>
> 1. **Appropriate Overestimation Mitigation**: Prevents Q-explosion as in Fig.2
> 2. **Generalization Safety**: Minimizes NTK-based error propagation (shown above)
>
> Thus,  distinguished from existing methods that impose explicit constraints on surface-level patterns, FANS **fundamentally mitigates distribution shifts by architecturally suppressing OOD overestimation and spurious generalization**—constituting our primary contribution.
>
> We provide both **theoretical grounding** (NTK dynamics) and **multi-faceted empirical evidence** (value accuracy + generalization metrics) to resolve concerns about FANS's advantages. **We sincerely hope this comprehensive response clarifies FANS's unique value. Given the novel evidence and analysis, we kindly request reconsidering the score to reflect these contributions. Thank you!**
>
> [1] Reining Generalization in Offline Reinforcement Learning via Representation Distinction. NeurIPS 2023.
>
> [2] DR3: Value-Based Deep Reinforcement Learning Requires Explicit Regularization. ICLR 2022.

---

> > ### Author Response · Authors · 2025-08-09
> >
> > Thank you very much for your time and thoughtful feedback on our manuscript. As the ddl of author-reviewer discussion phase is approaching, we would like to know whether these responses adequately address your questions. If any clarification is still needed, we would be happy to provide further details. Thank you again for your consideration, your insights are invaluable to us!

---

### Official Review · Reviewer_NuBS · 2025-07-01

**Clarity:** 2
**Significance:** 2
**Originality:** 3
**Rating:** 4
**Confidence:** 2

**Summary:**

The paper proposes FANS (Flatness-Aware Network Structure), a plug-in critic architecture for offline actor-critic RL that swaps the usual MLP for residual blocks with smooth Gaussian activations, LayerNorm, and a small ensemble.

**Questions:**

Please respond to the weakness described above. I am happy to raise my score based on the further discussions.

**Ethical Concerns:**

["NO or VERY MINOR ethics concerns only"]

**Final Justification:**

Thank you for the clarification. I would like to increase my score to 4.

**Limitations:**

Yes

**Quality:**

3

**Strengths And Weaknesses:**

## Strength

* **Architectural novelty** – FANS is the first offline-RL approach that combats OOD value over-estimation *solely* through a flatness-aware critic architecture (residual blocks + Gaussian activations + LayerNorm + small ensemble), in contrast to prior work that inserts explicit conservative penalties such as CQL and IQL.
* **Hyper-parameter-free simplicity** – because the vanilla TD3 loss functions remain unchanged, FANS eliminates the extra tuning knobs that regularizer methods require (e.g., CQL’s penalty weight, IQL’s expectile τ) and can be dropped into existing actor-critic pipelines without any objective-level modifications.
* **Strong, complementary performance** – on all nine D4RL MuJoCo tasks the plain TD3 + FANS agent surpasses state-of-the-art conservative baselines, and when FANS is combined with those same baselines it delivers further gains, showing both superior standalone effectiveness and orthogonality to regulariser techniques.

## Weaknesses

* **On motivation**

  I don't quite understand the motivation of the paper. While FANS clearly outperforms conservative-regularizer baselines, it is not obvious why an architectural change is the necessary remedy. Could the authors please spell out the fundamental limitations of regularizer-based methods (if any) and explain why these issues cannot be mitigated without altering the network architecture ? This would make the motivation much more compelling.

* **On conceptual framing**

  The paper’s central narrative is that FANS “steers optimization toward flatter minima,” which, in turn, mitigates OOD over-estimation. Yet the manuscript doesn't quantify flatness (e.g., Hessian trace) nor compares those metrics against regularizer baselines. As a result, how architecture impacts flatness and then further affects robustness remains unclear to me.

---

> ### Author Rebuttal · Authors · 2025-07-31
>
> Thank you for your thoughtful and inspiring comments, which will help us greatly in improving the quality of our work. We provide discussions and explanations about your concerns as follows.
>
> **> 1. Response to Motivation and the Role of Architecture:**
>
> We thank you for your insightful question. While regularizer-based methods (e.g., conservative Q-learning, behavior cloning constraints) have proven effective in offline RL by mitigating extrapolation errors and discouraging OOD actions, they primarily operate at the **objective level** by modifying the loss function. These methods aim to influence learning behavior via explicit penalties or constraints, often without considering the inductive biases introduced by the network architecture itself.
>
> However, a fundamental limitation of regularizer-based approaches is their inability to directly control the **geometry of the loss landscape** or the **sharpness of the learned solution**. In the offline RL setting, where learning relies heavily on bootstrapped targets and limited data coverage, sharp or brittle solutions can lead to unstable training and poor generalization. Regularizers may reduce certain types of estimation error but do not inherently promote **flatter minima** or **smoother optimization dynamics**.
>
> Our motivation for exploring an **architectural remedy** arises from this observation. The FANS framework is designed to structurally guide the optimization process toward **flatter and more robust regions** of the solution space. Components such as residual blocks, Gaussian activations, and layer normalization are known to promote gradient smoothness and stability, while lightweight ensemble modeling helps reduce estimation variance. These architectural choices introduce inductive biases that **complement, rather than replace,** regularization.
>
> Importantly, our goal is not to position architecture as a replacement for regularization-based methods, but rather as an **orthogonal and complementary** perspective. In principle, architectural and objective-level approaches can be combined. In this work, however, we isolate the architectural effects to highlight their **standalone benefits** and to motivate further exploration of **structure-aware design** in offline RL.
>
> We will clarify this motivation more explicitly in the revised manuscript.
>
> **> 2. Response to Quantifying Flatness and the Conceptual Framing:**
>
> We sincerely appreciate your insightful comments, which indeed highlight aspects not covered in the initially submitted manuscript. Thank you for bringing these to our attention. One of the core motivations of this work is to guide the optimization process toward **flatter minima** through **structural design**, thereby improving **generalization** in offline RL, especially under **distributional shift**. However, as the reviewer correctly noted, the current version of the paper does not explicitly include quantitative measures of flatness, such as the **Hessian trace**, **eigenvalues**, or **sharpness**, which is a limitation of our current work.
>
> We chose not to include such analysis in the main paper for the following reasons: First, in RL, especially when using **bootstrapped targets** like Q-value estimates, the **loss surface is highly non-stationary**, making the direct computation of the **Hessian** unstable and computationally expensive. Second, this work focuses on understanding the role of **architecture itself** by isolating it from objective-level interventions. Therefore, we primarily evaluated improvements in **OOD generalization**, **estimation variance**, and overall policy performance as **indirect evidence** of improved flatness and robustness.
>
> We fully agree that the causal link between **architectural design**, **loss surface flatness**, and **policy robustness** should be more rigorously demonstrated. Thus, in the revised version, we will supplement the paper with additional analyses to characterize the **flatness of the loss landscape**. Since we cannot include additional figures or appendices in the rebuttal, we have instead described our methodology for measuring landscape flatness in **Reviewer jCXx (#R2)’s comment 3** for reference.

---

> > ### Comment · Reviewer_NuBS · 2025-08-05
> > **Reply**
> >
> > Thank you for the clarification. I've increased my score to 4.

---

> > > ### Author Response · Authors · 2025-08-05
> > >
> > > Thank you so much for increasing your score! We're very happy to have addressed your concerns and will make sure to include the corresponding revisions in a future version of the paper!

---

### Official Review · Reviewer_jCXx · 2025-07-02

**Clarity:** 3
**Significance:** 3
**Originality:** 3
**Rating:** 5
**Confidence:** 4

**Summary:**

This work focuses on architectural design in offline reinforcement learning and introduces FANS (Flatness-Aware Network Structure) to optimize neural networks by leveraging the flatness of the loss landscape. The work proposes FANS to guide training toward flatter regions of the loss landscape, which often relate to better generalization performance. FANS combines residual blocks, Gaussian activation functions, layer normalization, and ensemble techniques to improve generalization. The paper provides empirical evidence demonstrating FANS's effectivness across various datasets.

**Questions:**

- How is the computational overhead of FANS?
- The empirical results primarily emphasize performance. Is there a way to demonstrate that FANS effectively guides training dynamics toward flatter regions of the loss landscape? I understand it is difficult to visualize the loss landscape in RL, but if such results exist, they would strengthen this paper.
- How many hyperparameters does the method introduce? How sensitive is the method to hyperparameter choices?

**Ethical Concerns:**

["NO or VERY MINOR ethics concerns only"]

**Final Justification:**

I thank the authors for their detailed response. The proposed method for measuring landscape flatness is reasonable, and the time complexity comparison is clear. I encourage the authors to include both the time complexity analysis and the loss landscape results in the revised version of the paper, as these additions would further enhance the completeness of the work. I think that FANS presents a simple yet effective architectural design for offline RL that achieves strong performance. Taking the rebuttal into account, I have updated my score accordingly.

**Limitations:**

A more explicit discussion of the study's limitations would further enhance the paper's academic rigor.

**Quality:**

3

**Strengths And Weaknesses:**

## Strengths
- Instead of typical algorithmic development, the paper introduces architectural design to enhance neural network generalizstion in offline reinforcement learning, exploring a relatively under-investigated area.
- The empirical performance of FANS is impressive, and ablation studies are included to confirm the effectiveness of each of the four components in FANS.
- The proposed method is compatible with various offline reinforcement learning algorithms and can be easily integrated into existing ones.
- The paper is well-written, with clear explanations of the method and results, making it easy to follow.

## Weaknesses
- In the limited data experiments (Figure 3), to test the generalization ability, FANS should be compared with other offline RL methods rather than the online method TD3, which is known to suffer in the offline setting.

---

> ### Author Rebuttal · Authors · 2025-07-31
>
> We extend our sincere gratitude for your inspiring and professional insights, which are invaluable in enhancing the quality of our work. In response to your concerns, we report below the explanations and answer your questions accordingly.
>
> **> 1. Response to Computational Overhead of FANS:**
>
> We provided a detailed analysis of FANS’s computational overhead in our **response to Reviewer RRU4 (#R1), point 2**. We hope this is helpful to you.
>
> **> 2. Response to Hyperparameter Count and Sensitivity:**
>
> **FANS introduces minimal additional hyperparameters compared to standard offline RL baselines**. Its architectural components are largely self-contained and do not require extensive tuning. The table below summarizes the key components introduced by FANS and whether they introduce new tunable hyperparameters.
>
> | **Component**        | **Hyperparameter**     | **Notes**                                                                 |
> |----------------------|------------------------|---------------------------------------------------------------------------|
> | Residual Block       | None                   | Structural replacement for MLP layers; no additional tunable parameters. |
> | Gaussian Activation  | None                   | Fixed as $f(x) = \exp(-x^2)$; no hyperparameter introduced.               |
> | Layer Normalization  | None                   | Standard layer-wise normalization; no tuning typically required.         |
> | Ensemble Modeling    | Number of models $M$   | Primary additional hyperparameter; typically $M=2\sim5$.                  |
>
> **FANS also exhibits low sensitivity to hyperparameter choices**. The use of residual connections and layer normalization enhances training stability and enables deeper architectures without introducing optimization instability. The fixed Gaussian activation promotes smoother gradients and helps reduce sensitivity to learning rates and batch sizes. Overall, FANS maintains robust performance across a broad range of hyperparameter settings, alleviating the need for exhaustive tuning.
>
> **> 3. Response to Flatter Regions of the Loss Landscape:**
>
> Your suggestion is highly valuable. Considering that we are unable to submit a PDF response, we outline below a proposed approach to illustrate whether FANS effectively guides training dynamics toward flatter regions of the loss landscape:
>
> (1) **Selection of Converged Policies:**
>    We identify a point in training where the loss stabilizes and performance metrics (e.g., policy returns) no longer change significantly, indicating convergence. At this stage, we fix the parameters $\phi$ of the policy network $\pi_\phi$.
>
> (2) **Local Surface Construction:**
>    We randomly sample a batch of states from the replay buffer and apply small perturbations to each state. For each perturbed state, we compute the corresponding action using the fixed policy network $\pi_\phi$. The original (unperturbed) state's action is treated as the reference optimal action.
>
> (3) **Surface Visualization:**
>    We construct a 2D surface where the *x*- and *y*-axes correspond to the direction and magnitude of perturbations in the state space, and the *z*-axis represents the normalized Euclidean distance between the perturbed action and the reference action. Concave points on this surface correspond to the optimal actions. A flatter surface indicates that the policy output is less sensitive to input perturbations, suggesting greater local robustness and a smoother loss landscape.
>
> (4) **Comparative Analysis:**
>    We perform this analysis on policies trained *with* and *without* FANS. By comparing the resulting action surfaces, we aim to visually demonstrate that FANS leads to a flatter, smoother local landscape $-$ supporting the claim that it enhances training stability and generalization.
>
> Although directly visualizing the high-dimensional loss landscape in reinforcement learning is inherently difficult, this approach serves as an interpretable proxy. We will include such comparative plots in the final version of the paper to strengthen the empirical support for FANS.
>
> **> 4. Response to Weakness: Inappropriate Baseline in Figure 3:**
>
> Thank you for your valuable feedback. Reviewer 6MWm (#R4) has raised a similar concern, and we kindly refer you to our detailed **response to #R4, point 1**, for your reference.

---

> > ### Comment · Reviewer_jCXx · 2025-08-05
> >
> > I thank the authors for their detailed response. The proposed method for measuring landscape flatness is reasonable, and the time complexity comparison is clear. I encourage the authors to include both the time complexity analysis and the loss landscape results in the revised version of the paper, as these additions would further enhance the completeness of the work. I think that FANS presents a simple yet effective architectural design for offline RL that achieves strong performance. Taking the rebuttal into account, I have updated my score accordingly.

---

> ### Author Response · Authors · 2025-08-06
>
> Thanks for your supportive feedback! We will include the supplymentary results in our revised paper according to your and other reviewers' suggestions to further enhance the completeness of the work.

---

### Official Review · Reviewer_RRU4 · 2025-07-03

**Clarity:** 3
**Significance:** 2
**Originality:** 3
**Rating:** 4
**Confidence:** 4

**Summary:**

This paper presents FANS,  a generalization-oriented structured network framework that promotes flatter and robust policy learning by guiding the optimization trajectory through modular architectural design. FANS comprises four key components: (1) Residual Blocks, (2) Gaussian Activation, (3) Layer Normalization,  and (4) Ensemble Modeling.

**Questions:**

* FANS includes four key modules, but I couldn’t find any ablation study that shows which parts contribute the most. Could the authors provide insights or experiments on the relative importance of each component?
* Since a new network architecture is introduced, it would be helpful to analyze or report how its training cost compares with previous offline RL baselines. This would provide insight into the practical efficiency of the proposed approach.
* The comparison baselines are not the SOTA methods. Can you add more experiments with recent offline RL methods? such as FQL, A2PR, SORL. Moreover, the related work section does not discuss the recent methods mentioned above. Could you consider adding some discussion on these newer approaches?

Reference:

Park, S., Li, Q., & Levine, S.. Flow Q-Learning. arXiv preprint arXiv:2502.02538.

Liu, T., Li, Y., Lan, Y., Gao, H., Pan, W., & Xu, X.. Adaptive Advantage-Guided Policy Regularization for Offline Reinforcement Learning. In International Conference on Machine Learning (pp. 31406-31424). PMLR.

Espinosa-Dice, N., Zhang, Y., Chen, Y., Guo, B., Oertell, O., Swamy, G., ... & Sun, W.. Scaling Offline RL via Efficient and Expressive Shortcut Models. arXiv preprint arXiv:2505.22866.

**Ethical Concerns:**

["NO or VERY MINOR ethics concerns only"]

**Final Justification:**

We sincerely appreciate the authors for providing the additional experimental results during the rebuttal. These results help clarify the effectiveness of the proposed method and support some of the claims made in the paper. Based on the clarifications and additional experiments provided during the rebuttal, I will update my score accordingly.

**Limitations:**

* Lack of ablation studies to analyze the contribution of each component.

* Missing experimental comparison with state-of-the-art (SOTA) methods.

* Insufficient main experimental results to fully validate the effectiveness of the proposed approach.

**Quality:**

3

**Strengths And Weaknesses:**

### Strengths
* FANS can easily plug in a standard actor-critic framework.
* This remarkably simple architecture of FANS achieves superior performance across various tasks.
### Weaknesses
* This paper lacks some experimental comparison with the SOTA methods.
* The experimental evaluation is limited, as it only includes D4RL MuJoCo tasks. The performance of the proposed method on more challenging environments such as AntMaze or Adroit remains unclear.

---

> ### Author Rebuttal · Authors · 2025-07-31
>
> We sincerely appreciate your thoughtful and professional insights, which have greatly improved the quality of our work. In response to your concerns, we provide the explanations and answers below.
>
> **> 1. Response to Lack of Ablation Studies:**
>
> We have **conducted ablation studies in Section 4.4** that systematically evaluate the contribution of each module within FANS to the overall performance. Additionally, **Appendix B.2 provides further structural analysis**, verifying the effectiveness of each component through parameter variations. For **a more detailed explanation of the ablation experiments, please also refer to our response to Reviewer 6MWm (#R4), point 3**. We hope these clarifications help to better illustrate the relative importance of the components.
>
> **> 2. Response to the Time Complexity and Computational Overhead of FANS:**
>
> **(1) Residual Block Complexity:**
>
> Each residual block in FANS (as shown in Table 1 of the main paper) comprises two linear layers, a LayerNorm, a Gaussian activation, and a residual connection.
>
> - The dominant operations are the two linear layers, each with a time complexity of $\mathcal{O}(d^2)$, where $d$ is the hidden dimension.
> - LayerNorm, the Gaussian activation ($\exp(-x^2)$), and the residual addition are all element-wise operations with complexity $\mathcal{O}(d)$.
>
> Thus, each block maintains an overall complexity of **$\mathcal{O}(d^2)$**, consistent with standard MLP layers used in typical offline RL networks. The additional operations increase the constant factor but do not alter the asymptotic complexity. Additionally, these Residual Blocks are only applied to the critic network, remaining lightweight and efficient.
>
> **(2) Ensemble Modeling:**
>
> We acknowledge that ensemble modeling introduces additional cost, scaling linearly with the number of models $M$, resulting in **$\mathcal{O}(M \cdot d^2)$** complexity. However:
>
> - The ensemble is only used in the critic, not the actor;
> - We adopt a small $M$ (e.g., 2-5) while still achieving strong performance $-$ significantly fewer than SAC-N, which requires 100-500 Q-networks, and EDAC, which requires 20-50 Q-networks;
>
> **(3) Practical Impact:**
>
> Despite the slight increase in per-iteration cost, FANS demonstrates stronger generalization and stability, requiring fewer training epochs overall. In Figure 6 of the Appendix, we show that FANS converges faster than strong baselines under the same compute budget.
>
> **Summary:**
>
> In summary, while FANS introduces moderate constant-factor overhead due to its structured architectural design, the **overall time complexity remains $\mathcal{O}(d^2)$**, matching conventional offline RL networks. The improvements in generalization and training robustness justify this trade-off. Bellow the table "Time Complexity and GPU Consumption Comparison":
>
> | **Method**     | **Critic Architecture**                                         | **#Q Networks** | **Per-Step Time Complexity**     | **GPU Memory Footprint** | **Notes**                                                                 |
> |----------------|------------------------------------------------------------------|------------------|----------------------------------|---------------------------|----------------------------------------------------------------------------|
> | **CQL**        | Standard MLP                                                    | 2                | $\mathcal{O}(d^2)$               | ★☆☆☆☆                    | No ensemble; low cost but prone to overestimation                         |
> | **EDAC**       | Ensemble + Gradient Decorrelation                               | 20–50            | $\mathcal{O}(M \cdot d^2)$       | ★★★★☆                    | Strong baseline but expensive ensemble                                    |
> | **SAC-N**      | Massive Ensemble                                                | 100–500          | $\mathcal{O}(M \cdot d^2)$       | ★★★★★                    | High Q-network redundancy, heavy computational burden                     |
> | **FANS (Ours)**| Residual Blocks + Gaussian Act + LayerNorm + Small Ensemble     | 2–5              | $\mathcal{O}(d^2)$ + $\mathcal{O}(M \cdot d^2)$ | ★★☆☆☆         | Structured critic only; lightweight design; converges faster              |
>
> We will include detailed time complexity analysis and empirical runtime comparisons in the revised version.
>
> **> 3. Response to Missing Experimental Comparison with state-of-the-art (SOTA) Methods:**
>
> Thank you for the valuable suggestion. As for A2PR, we directly extracted their MuJoCo Gym results from their original paper. The comparison between FANS and A2PR is summarized in the table below, where FANS outperforms A2PR on 7 out of 9 tasks. We will include this comparison in the revised manuscript’s main text.
>
> |               | ha-m           | ho-m            | wa-m           | ha-mr          | ho-mr           | wa-mr          | ha-me          | ho-me           | wa-me           |
> |---------------|----------------|-----------------|----------------|----------------|-----------------|----------------|----------------|-----------------|-----------------|
> | A2PR          | **68.61±0.37** | 100.79±0.32    | 89.73±0.60     | **56.58±1.33** | 101.54±0.90    | 94.42±1.54    | 98.25±3.20     | 112.11±0.32    | 114.62±0.78    |
> | **TD3+FANS**    | 66.6±0.8      | **104.6±0.9**   | **101.0±1.6**  | 55.9±1.5       | **103.2±1.1**   | **98.3±2.0**   | **102.8±3.4**  | **113.3±1.4**   | **118.1±0.4**   |
>
> Regarding FQL and SORL, we note that neither method reports results on the standard MuJoCo benchmark, which serves as the primary evaluation suite in our study. As a result, a direct empirical comparison was not feasible under our current experimental setup. Nevertheless, we greatly value these contributions and will explicitly include a discussion of both methods in the Related Work section of the revised version, clarifying how they relate to our approach. We are grateful to you for pointing out these important works.
>
> **> 4. Response to Insufficient Main Experimental Results:**
>
> Due to the limited time available during the rebuttal phase, we have conducted a preliminary set of experiments on the AntMaze and Adroit tasks. The results are shown in the table below. More comprehensive results will be provided in the revised version of the paper.
>
> **(1) AntMaze:**
>
> |               | antmaze-umaze | antmaze-umaze-diverse | antmaze-medium-play | antmaze-medium-diverse | antmaze-large-play | antmaze-large-diverse |
> |---------------|---------------|-----------------------|---------------------|------------------------|--------------------|-----------------------|
> | TD3+BC         | 70.75±39.18 | 44.75±11.61         | 0.25±0.43         | 0.25±0.43            | 0.00±0.00        | 0.00±0.00           |
> | **TD3+FANS**          | **87.75±10.48**  | **72.25±7.54**           | **3.25±1.30**          | **5.75±0.70**             | 0.00±0.00        | 0.00±0.00           |
>
> **(2) Adroit:**
>
> |               | pen-human  | pen-cloned | door-human | door-cloned | hammer-human | hammer-cloned | relocate-human | relocate-cloned |
> |---------------|---------------|-----------------------|---------------------|------------------------|--------------------|-----------------------|--------------------|-----------------------|
> | TD3+BC         | -3.88±0.21 | 5.13±5.28    | -0.33±0.01   | -0.34±0.01  | 1.02±0.24       | 0.25±0.01         |-0.29±0.01      | -0.30±0.01         |
> | **TD3+FANS**          | **55.16±5.49**  | **70.43±6.31**     | **9.72±6.13**       | **10.37±6.73**             | **2.24±0.18**        | **1.52±0.38**           |**0.10±0.09**        | **0.17±0.59**  |

---

> > ### Comment · Reviewer_RRU4 · 2025-08-04
> > **Rebuttal by Reviewer RRU4**
> >
> > We sincerely appreciate the authors for providing the additional experiments. We also encourage the authors to include the relevant ablation studies and main comparisons in the paper, as doing so would further improve the quality and completeness of the work. Based on the new results and clarifications provided, I will revise my score accordingly.

---

> > > ### Author Response · Authors · 2025-08-05
> > >
> > > Thanks for your advices and the score revision! We'll include the supplymentary results in the paper!

---

### Note · Authors · 2025-08-12

We thank the reviewers and the AC for their time and constructive feedback. Following the rebuttal phase, **three reviewers (RRU4, jCXx, NuBS) have acknowledged our clarifications and increased their scores**, recognizing the novelty and empirical strength of our work.

Regarding the remaining concern from Reviewer 6MWm about "Baselines" and "Q-value Overestimation", we provide both **theoretical grounding (NTK dynamics)** and multi-faceted **empirical evidence (value accuracy + generalization metrics)** to resolve concerns about FANS's advantages in ***Further Response to Reviewer 6MWm (PART 1 and PART 2)***. These clarifications comprehensively address the raised issues, and we believe they fully resolve the reviewer's concern while reinforcing FANS's unique advantages.

With all major concerns addressed, the final version of our work presents a well-substantiated and rigorously evaluated contribution that will be of broad interest and lasting value to the offline RL community. We appreciate the reviewers' and AC's efforts, and are confident that the clarified contributions and results demonstrate the novelty, rigor, and significance of our work.

---

### Decision · Program_Chairs · 2025-09-17

**Decision:**

Accept (poster)

**Comment:**

This paper proposes a network architecture framework for improved out of data generalization in offline reinforcement learning. Unlike other offline RL papers, this paper focuses on new network architectures allowing the method to be easily integrated into standard actor-critic methods without altering the loss function. Empirical results on D4RL benchmark show that TD3 + FANS achieves consistent improvements over other baselines.

Strengths:

The focus of the paper is not to propose another offline RL loss function but rather simple architecture change.

The method achieves strong performances with a lot less Q network ensembles with only 2-5 Q networks compared to 20-25 in EDAC.

Weaknesses:

Flatness claims are primarily speculative and has no theoretical insight.

Experiments are reported only on D4RL benchmark. During the rebuttal additional experiments were added on more challenging AntMaze/Adriot environments.

During the rebuttal, the authors provided ablation studies clarifying the role of each component, extended comparison to newer baselines, additional results on more challenging environments, detailed analysis of computational efficiency. Importantly, the authors demonstrated that FANS achieves comparable or better generalization with 2–5 Q-networks versus 20–50 in EDAC, highlighting its efficiency. They also provided Q-value over/underestimation experiments and NTK-based generalization analyses. Several reviewers raised their scores after these clarifications, and consensus moved toward acceptance.

The decision is to accept. The proposed framework offers a simple yet effective architectural contribution for offline RL with strong empirical support. For the camera ready version, I strongly encourage the authors to 1) include the Q-value over/underestimation results, NTK analysis presented during rebuttal, 2) expand the experimental results for AntMaze/Adroit, 3) more explicitly connect architectural design to flatness claims.